# Adversarial Attacks on Online Learning to Rank with Stochastic Click Models

**Zichen Wang**                                                                    *zichenw6@illinois.edu*
*Department of Electrical Computer Engineering & Coordinated Science Laboratory*
*University of Illinois Urbana-Champaign*

**Rishab Balasubramanian**                                                         *balasuri@oregonstate.edu*
*Department of Computer Science*
*Oregon State University*

**Yuan Hui**                                                                       *huiyuan@princeton.edu*
*Department of Electrical Computer Engineering*
*Princeton University*

**Chenyu Song**                                                                    *songchen@oregonstate.edu*
*Department of Computer Science*
*Oregon State University*

**Mengdi Wang**                                                                    *mengdiw@princeton.edu*
*Department of Electrical Computer Engineering*
*Princeton University*

**Huazheng Wang**                                                                  *huazheng.wang@oregonstate.edu*
*Department of Computer Science*
*Oregon State University*

**Reviewed on OpenReview:** *https://openreview.net/forum?id=BKwGowROBt*

## Abstract

We propose the first study of adversarial attacks on online learning to rank. The goal of the attacker is to misguide the online learning to rank algorithm to place the target item on top of the ranking list linear times to time horizon $T$ with a sublinear attack cost. We propose generalized list poisoning attacks that perturb the ranking list presented to the user. This strategy can efficiently attack any no-regret ranker in general stochastic click models. Furthermore, we propose a click poisoning-based strategy named attack-then-quit that can efficiently attack two representative OLTR algorithms for stochastic click models. We theoretically analyze the success and cost upper bound of the two proposed methods. Experimental results based on synthetic and real-world data further validate the effectiveness and cost-efficiency of the proposed attack strategies.

## 1 Introduction

Online learning to rank (OLTR) (Grotov and de Rijke, 2016) formulates learning to rank (Liu et al., 2009), the core problem in information retrieval, as a sequential decision-making problem. OLTR is a family of online learning solutions that exploit implicit feedback from users (e.g., clicks) to directly optimize parameterized rankers on the fly. It has drawn increasing attention in recent years (Kveton et al., 2015a; Zoghi et al., 2017; Lattimore et al., 2018; Oosterhuis and de Rijke, 2018; Wang et al., 2019; Jia et al., 2021; Wang et al., 2018; Ai et al., 2021) due to its advantages over traditional offline learning-based solutions and numerous applications in web search and recommender systems (Liu et al., 2009).

To effectively utilize users' click feedback to improve the quality of ranked lists, one line of OLTR studied bandit-based algorithms under different click models. In each iteration, the algorithm presents a ranked list of $K$ items selected from $L$ candidates based on its estimation of the user's interests. The ranker observes the user's click feedback and updates these estimates accordingly. Different users may examine and click on the ranking list differently, and how the user interacts with the item list is called the *click model*. Many works have been dedicated to establishing OLTR algorithms in the cascade model (Kveton et al., 2015a;b; Zong et al., 2016; Li et al., 2016; Vial et al., 2022), the position-based model (Lagrée and Vernade, 2016) and the dependent click model (Katariya et al., 2016; Liu et al., 2018). However, these algorithms are ineffective when employed under a different click model. To overcome this bottleneck, Zoghi et al. (2017); Lattimore et al. (2018); Li et al. (2019) proposed OLTR algorithms with general stochastic click models that cover the aforementioned click models.

There has been a huge interest in developing robust and trustworthy information retrieval systems (Golrezaei et al., 2021; Ouni et al., 2022; Sun and Jafar, 2016), and understanding the vulnerability of OLTR algorithms to adversarial attacks is an essential step towards the goal. Recently, several works explored adversarial attacks on multi-armed bandits (Jun et al., 2018; Liu and Shroff, 2019) and linear bandits (Garcelon et al., 2020; Wang et al., 2022) where the system recommends one item to the user in each round. The idea of the poisoning attack is to lower the rewards of the non-target item to misguide the bandit algorithm to recommend the target item using cost sublinear to time horizon $T$. In online ranking, we consider the goal of the adversary as misguiding the algorithm to rank the target item on top of the ranking list linear times $(T - o(T))$ with sublinear attack cost $(o(T))$. However, it is hard to directly extend the attack strategy on multi-armed bandits to OLTR since the click model is a black box to the adversary.

In this paper, we propose the first study of adversarial attacks on OLTR to misguide the rankers to place the target item on top of the ranking list. We study two threat models: click poisoning attacks where the adversary manipulates the rewards the user sends back to the ranking algorithm, and list poisoning attacks where the adversary perturbs the ranking list that are presented to the user. We first propose a generalized list poisoning attack strategy (`GA`) that can *efficiently attack any no-regret ranker* for stochastic click models. The adversary perturbs the ranking list presented to the user and pretends the click feedback represents the user's interests in the original ranking list. This guarantees the feedback always follows the unknown click model, making the attack *stealthy*. Furthermore, we propose a click poisoning-based strategy named attack-then-quit strategy (`ATQ`) that can efficiently attack two representative OLTR algorithms for *stochastic click models*, i.e., BatchRank (Zoghi et al., 2017) and TopRank (Lattimore et al., 2018). Our theoretical analysis guarantees that the proposed methods succeed with sublinear attack cost. We empirically evaluate the proposed methods against several OLTR algorithms on synthetic data and a real-world dataset under different click models. Our experimental results validated the theoretical analysis of the effectiveness and cost-efficiency of the two proposed attack algorithms. Our code and data can be accessed publicly for reproducibility.[1].

## 2 Related Works

**Online learning to rank.** OLTR is first studied as ranked bandits (Radlinski et al., 2008; Slivkins et al., 2013), where each position in the list is modeled as an individual multi-armed bandits problem (Auer et al., 2002). Such a problem can be settled down by bandit algorithms which can maximize the expected click number in each round. Recently studied of OLTR focused on different click models (Craswell et al., 2008; Chuklin et al., 2015), including the cascade model (Kveton et al., 2015a;b; Zong et al., 2016; Li et al., 2016; Vial et al., 2022), the position-based model (Lagrée and Vernade, 2016) and the dependent click model (Katariya et al., 2016; Liu et al., 2018). OLTR with general stochastic click models is studied in (Zoghi et al., 2017; Lattimore et al., 2018; Li et al., 2018; 2019; Gauthier et al., 2022).

**Adversarial attack against bandits.** Adversarial reward poisoning attacks against multi-armed bandits have been recently studied in stochastic bandits (Jun et al., 2018; Liu and Shroff, 2019; Zuo, 2020; Xu et al., 2021; Lu et al., 2021), linear bandits (Wang et al., 2022; Garcelon et al., 2020), Gaussian process bandits (Han and Scarlett, 2021), adversarial bandits (Ma and Zhou, 2023), and combinatorial bandits (Balasubramanian

---

[1] https://github.com/rishabbala/Online-Learning-to-Rank-for-Stochastic-Click-Models

et al., 2023). These works share a similar attack idea, where the attacker holds the reward of the target arm, meanwhile lowers the reward of the non-target arm. Besides reward poisoning attacks, other threat models such as action poisoning attacks (Liu and Lai, 2020a; 2021) were also being studied.

Recently, Zuo et al. (2023) studied adversarial attacks on OLTR. The paper considered an *easier* attack goal: ensuring that the target item appears on the displayed length-$K$ ranking list $\mathcal{R}_t$ for $T - o(T)$ times, without regard for the specific position of the target item. They introduced two click poisoning attack algorithms specifically designed for CascadeUCB and PBMUCB. These algorithms rely on the knowledge of the target rankers and the underlying click model (i.e., Cascade model and position-based model). Additionally, they proposed a general attack algorithm based on click poisoning, operating under the assumption that the attacker is aware of the feasible feedback space of the victim's click model (refer to the last paragraph of page 8 of Zuo et al. (2023) for details). The authors acknowledged that "without this knowledge, ensuring valid post-attack feedback becomes impossible." Moreover, we illustrate through Example 2 that click poisoning attacks are poorly adapted to different click models. This discussion highlights the ineffectiveness of their attack algorithms (two single-target attack algorithms and a general attack algorithm) when information about the underlying click model is absent.

In this paper, we focus on fooling the victim ranker into placing the target item at the *top* of the list $\mathcal{R}_t$ for $T - o(T)$ times. To overcome the strong assumption that the attacker knows the underlying click model, we propose the list poisoning-based `GA` algorithm, which achieves our attack objective without relying on any knowledge of the underlying click model. We also introduce a new click poisoning-based attack strategy named `ATQ` to boost the target item to the top of the list. This strategy leverages the phase elimination property of general rankers like TopRank and BatchRank. We observe that these algorithms tend to "assume" the most attractive item is the best during certain exploratory phases, consistently placing it at the top. Consequently, our `ATQ` aims to fool TopRank and BatchRank into believing that the target item possesses the highest attractiveness within a short period.

## 3 Preliminaries

### 3.1 Online Learning to Rank

We denote the total item set with $L$ items as $\mathcal{D} = \{a_1, ..., a_L\}$. Let $\Pi_K(\mathcal{D}) \subset \mathcal{D}^K$ stands for all $K$-tuples with different elements from $\mathcal{D}$. At each round $t$, the ranker would present a length-$K$ ordered list $\mathcal{R}_t = (\mathbf{a}_1^t, ..., \mathbf{a}_K^t) \in \Pi_K(\mathcal{D})$ to the user, where $\mathbf{a}_k^t$ is the item placed at the $k$-th position of $\mathcal{R}_t$. Generally, $K$ is a constant much smaller than $L$. When the user observes the provided list, he/she returns click feedback $\mathcal{C}_t = (\mathcal{C}_1^t, ..., \mathcal{C}_L^t)$ to the ranker where $\mathcal{C}_k^t = 1$ stands for user click on item $a_k$. Note that $a_k \notin \mathcal{R}_t$ can not be observed by the user, thus its click feedback in round $t$ is $\mathcal{C}_k^t = 0$. The attractiveness score represents the probability that the user is interested in item $a_k$, and it is defined as $\alpha(a_k) \in [0, 1]$, which is unknown to the ranker. Without loss of generality, we suppose $\alpha(a_1) >, ..., > \alpha(a_L)$ where $a_1$ is the most attractive item and $a_L$ is the least attractive item.

### 3.2 Stochastic Click Models

In this paper, we consider the general stochastic click models studied by Zoghi et al. (2017); Lattimore et al. (2018), where the conditional probability that the user clicks on position $k$ in round $t$ is only related to $\mathcal{R}_t$. This implies there exists an unknown function that satisfies

$$P(\mathcal{C}_s^t = 1 \mid \mathcal{R}_t = \mathcal{R}, \ \mathbf{a}_k^t = a_s) = v(\mathcal{R}, \mathbf{a}_k^t, k), \tag{1}$$

where $\mathcal{R} = \mathcal{R}_t$ denote the list presented in round $t$, and $a_s = \alpha_k^t$ denote that the $s$-th most attractive item (where $a_s$ can be any arbitrary item in $\mathcal{D}$) is placed in the $k$-th position of $\mathcal{R}_t$ (also the $k$-th position of $\mathcal{R}$). The key problem of OLTR is to present the optimal list $\mathcal{R}^* = (a_1, ..., a_K)$ to the user for per-round click number maximization. The optimal list is unique due to the attractiveness of items is unique.

**Assumption 1.** *Since the user does not observe items in position $\notin \mathcal{R}_t$, we assume that the ranker can achieve maximum expected number of clicks in round t if and only if $\mathcal{R}_t = \mathcal{R}^*$, i.e.*

$$\max_{\mathcal{R} \in \Pi_K(\mathcal{D})} \sum_{k=1}^{K} v(\mathcal{R}, \mathbf{a}_k^t, k) = \sum_{k=1}^{K} v(\mathcal{R}^*, \mathbf{a}_k^t, k). \tag{2}$$

**Definition 1** (Cumulative regret)**.** *The performance of a ranker can be evaluated by the cumulative regret, defined as*

$$R(T) = \mathbb{E}\left[T \sum_{k=1}^{K} v(\mathcal{R}^*, \mathbf{a}_k^t, k) - \sum_{t=1}^{T} \sum_{k=1}^{K} v(\mathcal{R}_t, \mathbf{a}_k^t, k)\right].$$

*Note that if Assumption 1 holds, $\mathcal{R}^*$ can uniquely maximize $\sum_{k=1}^{K} v(\mathcal{R}_t, \mathbf{a}_k^t, k)$, and every $\mathcal{R}_t \neq \mathcal{R}^*$ leads to non-zero regret.*

We present two classic click models (Chuklin et al., 2015; Richardson et al., 2007; Craswell et al., 2008) that are special instances of the stochastic click models.

**Position-based model.** The position-based model (Richardson et al., 2007) assumes the examination probability of the $k$-th position in list $\mathcal{R}_t$ is a constant $\chi(k) \in [0, 1]$. In each round, the user receives the ordered list $\mathcal{R}_t$. He/she would examine position $k$ with probability $\chi(k)$. If position $k$ is examined then the user would click item $\mathbf{a}_k^t$ with probability $\alpha(\mathbf{a}_k^t)$. Hence, the probability of item $\mathbf{a}_k^t$ is clicked by the user is

$$v(\mathcal{R}_t, \mathbf{a}_k^t, k) = \chi(k)\alpha(\mathbf{a}_k^t). \tag{3}$$

Note that the examination probability of items not in $\mathcal{R}_t$ is 0. Hence, the expected number of clicks in round $t$ is

$$\sum_{k=1}^{K} v(\mathcal{R}_t, \mathbf{a}_k^t, k) = \sum_{k=1}^{K} \chi(k)\alpha(\mathbf{a}_k^t). \tag{4}$$

The examination probabilities of the first $K$ positions are assumed to follow $\chi(1) > ... > \chi(K)$ (Chuklin et al., 2015). The maximum number of clicks in each round is $K$.

**Cascade model.** In the cascade model (Craswell et al., 2008), the user examines the items in $\mathcal{R}_t$ sequentially from $\mathbf{a}_1^t$. The user continues examining items until they find an item $\mathbf{a}_k^t$ attractive or they reach the end of the list. If the user finds $\mathbf{a}_k^t$ attractive, they would click on it and stop examining further.

According to the above description, the examination probability of position $k$ equals the probability of none of the items in the first $k - 1$ positions in $\mathcal{R}_t$ can attract the user, and can be represented as

$$\chi(\mathcal{R}_t, k) = \prod_{s=1}^{k-1}(1 - \alpha(\mathbf{a}_s^t)). \tag{5}$$

The maximum number of clicks is at most 1, and the expected number of clicks in each round can be written as

$$\sum_{k=1}^{K} v(\mathcal{R}_t, \mathbf{a}_k^t, k) = \sum_{k=1}^{K} \chi(\mathcal{R}_t, k)\alpha(\mathbf{a}_k^t) = 1 - \prod_{k=1}^{K}(1 - \alpha(\mathbf{a}_k^t)). \tag{6}$$

Similar to the position-based model, $\chi(\mathcal{R}_t, 1) > ... > \chi(\mathcal{R}_t, K)$ is hold in the cascade model.

**Definition 2** (No-regret ranker)**.** *We define the no-regret ranker as a ranker that achieves a sublinear ($o(T)$) regret in its click model under Assumption 1. By Definition 1, we can see that a ranker is no-regret if and only if it presents $\mathcal{R}^*$ to the user for $T - o(T)$ times.*

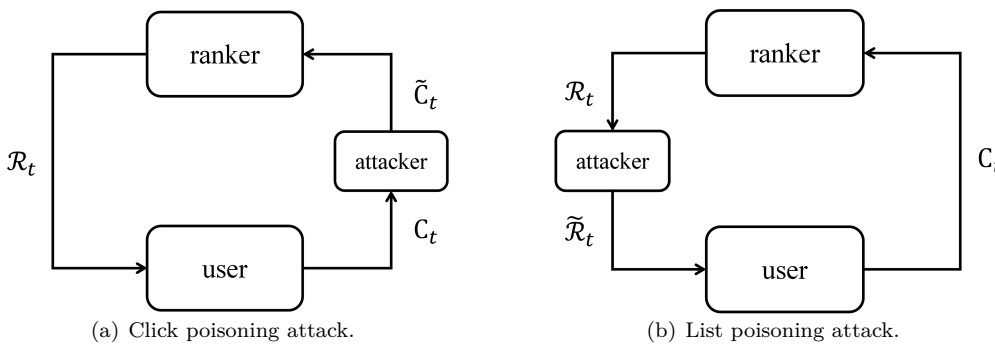

(a) Click poisoning attack.
(b) List poisoning attack.

Figure 1: Threat models on online learning to rank.

**Remark 1.** *We now briefly discuss relationship between click models and no-regret rankers. Recall the definition of the position-based model, the optimal list $\mathcal{R}^*$ can uniquely maximize (4). Thus, every ranker that achieves regret $R(T) = o(T)$ in the position-based model falls into the category of no-regret ranker (such as PBM-UCB (Lagrée and Vernade, 2016)). Besides, algorithms (i.e., BatchRank and TopRank) for general click models (Zoghi et al., 2017; Lattimore et al., 2018) present the optimal list $\mathcal{R}^*$ for $T - o(T)$ times owing to their elimination and divide-and-conquer nature. Therefore, they can achieve a sublinear regret under Assumption 1 and state-of-the-art online ranking methods BatchRank (Zoghi et al., 2017) and TopRank (Lattimore et al., 2018) fall into the category of no-regret rankers. However, every permutation of the first $K$-most attractive items can maximize Eq equation 6 in the cascade model. The item with the highest attractiveness may not be placed at the first position for $T - o(T)$ times by an online stochastic ranker with $R(T) = o(T)$. Thus not all rankers that achieve $R(T) = o(T)$ in the cascade model are no-regret rankers.*

### 3.3 Threat Models

Let $N_T(a_k) = \sum_{t=1}^{T} \mathbb{1}\{\alpha_1^t = a_k\}$ denote the total rounds item $a_k$ placed at the first position of $\mathcal{R}_t$ until time $T$. The adversary aims to fool the ranker to place a target item $\tilde{a}$ at the first position of $\mathcal{R}_t$ for $T - o(T)$ rounds. We consider two poisoning attack models.

**Click poisoning attacks.** We illustrate click poisoning attacks in Figure 1(a). This is similar to the reward poisoning attacks studied on multi-armed bandits (Jun et al., 2018; Liu and Shroff, 2019). In each round, the attacker obtains the user's feedback $\mathcal{C}_t$, and modifies it to perturbed clicks $\tilde{\mathcal{C}}_t = (\tilde{\mathcal{C}}_1^t, ..., \tilde{\mathcal{C}}_L^t)$. Naturally, the attacker needs to attain its attack goal with minimum attack cost defined as $\mathcal{C}(T) = \sum_{t=1}^{T} \sum_{k=1}^{L} |\tilde{\mathcal{C}}_k^t - \mathcal{C}_k^t|$.

**List poisoning attack.** Instead of directly manipulating the click feedback, the list poisoning attacks manipulate the presented ranking list from $\mathcal{R}_t$ to $\tilde{\mathcal{R}}_t$ as illustrated in Figure 1(b). This is similar to the action poisoning attack proposed by Liu and Lai (2020a; 2021) against multi-armed bandits. We assume the attacker can access items with low attractiveness denoted as $\{\eta_k\}_{k=1}^{2K-1} \notin \mathcal{D}$ and for convenience, $\alpha(\eta_1) >, ..., > \alpha(\eta_{2K-1})$. The low attractiveness items satisfy $\alpha(\eta_1) < \alpha(a_L)$. We suppose the attacker does not know the actual attractiveness of these items, but only their relative utilities, i.e., the attractiveness of items in $\{\eta_k\}_{k=1}^{K-1}$ is larger than items in $\{\eta_k\}_{k=K}^{2K-1}$. The attacker uploads these items to the candidate action set before exploration and we denote $\tilde{\mathcal{D}} = \mathcal{D} \cup \{\eta_k\}_{k=1}^{2K-1}$. In each round, the attacker can replace items in original ranking $\mathcal{R}_t$ with items in $\{\eta_k\}_{k=1}^{2K-1}$. This modified list $\tilde{\mathcal{R}}_t = (\tilde{\mathbf{a}}_1^t, ..., \tilde{\mathbf{a}}_K^t)$ is then sent to the user. The cost of the attack is $\mathcal{C}(T) = \sum_{t=1}^{T} \sum_{k=1}^{K} \mathbb{1}\{\tilde{\mathbf{a}}_k^t \neq \mathbf{a}_k^t\}$. Note that the click feedback $\mathcal{C}_t$ in list poisoning attacks is generated by $\tilde{\mathcal{R}}_t$ instead of $\mathcal{R}_t$, but the ranker assumes that the feedback is for $\mathcal{R}_t$.

The click poisoning attack has been studied in WSJ (2018); BuzzFeed (2019); Golrezaei et al. (2021). In practice, the click feedback manipulation can be executed through malware installed as a browser extension, which manipulates the click feedback signal locally before uploading it to the server (e.g., the OLTR algorithm). We also provide an example to explain the practicability of the list poisoning attack.

**Example 1** (Motivated example of list poisoning attack)**.** *Consider the OLTR of an e-commerce search engine (e.g., Amazon) where the attacker is a seller who wants to promote its target item (e.g., good) to the top (WSJ, 2018; BuzzFeed, 2019). The item set of the e-commerce consists of items uploaded by sellers.*

---

**Algorithm 1** Generalized List Poisoning Attack (`GA`)

---

1: **Inputs:** List $\mathcal{T} = (\tilde{a}, \eta_1, ..., \eta_{K-1})$ and $\{\eta_k\}_{k=1}^{2K-1}$
2: Upload $\{\eta_k\}_{k=1}^{2K-1}$ to the candidate action set
3: **for** $t = 1 : T$ **do**
4:     Observe $\mathcal{R}_t = (\mathbf{a}_1^t, ..., \mathbf{a}_K^t)$
5:     **if** $\mathcal{R}_t \backslash \mathcal{T} \neq \emptyset$ **then**
6:         **for** $k = 1 : K$ **do**
7:             **if** $\mathbf{a}_k^t \notin \mathcal{T}$ **then**
8:                 Set $\tilde{\mathbf{a}}_k^t = \eta_{K+k-1}$.
9:             **else**
10:                Set $\tilde{\mathbf{a}}_k^t = \mathbf{a}_k^t$
11:         Return $\tilde{\mathcal{R}}_t = (\tilde{\mathbf{a}}_1^t, ..., \tilde{\mathbf{a}}_K^t)$ to the user
12:     **else**
13:         Do not attack

---

*The attacker can find $2K - 1$ items with low attractiveness (e.g., unpopular advertisements or low-quality products) and derive their relative utilities (the relative utilities of $\eta_i$ between $i \in [1, K-1]$ and $i \in [K, 2K-1]$ ) based on offline learning to rank techniques. Due to low-attractiveness items being much more common in real-world applications compared to attractive ones and being much smaller than the total item number $L$, we believe finding $2K - 1$ low-attractiveness items is applicable. As a seller, the attacker could upload both low-attractive items and the target item to the item set (e.g., a seller listing products on the Amazon platform). Additionally, list manipulation can be executed through malware installed as a browser extension, which locally alters the ranking list on the web page. When the e-commerce platform interacts with the user, the attacker can implement a list poisoning attack strategy using the uploaded items and the malware.*

We define the *efficient attack strategies* as follows.

**Definition 3** (Efficient attack)**.** *We say an attack strategy is efficient if*

    *1. It misguides an online stochastic ranker to place the target item $\tilde{a}$ at the first position of $\mathcal{R}_t$ for $T - o(T)$ times in expectation with cost $\mathcal{C}(T) = o(T)$.*

    *2. To keep the click poisoning attack stealthy, the returned total clicks $\sum_{k=1}^{L} \tilde{\mathcal{C}}_k^t$ in the cascade model is at most 1 and in the position-based model is at most $K$.*

We conclude the preliminary with the difference between poisoning attacks on stochastic bandits (Jun et al., 2018; Liu and Shroff, 2019; Xu et al., 2021) and online learning rankers. Data poisoning attack on stochastic bandits aims to fool the bandit algorithm to pull the target arm $T - o(T)$ times with $o(T)$ cost. The main idea of this class of attack strategies is to hold the expected reward of the target item and reduce the expected reward of the non-target items. However, in the OLTR setting, 1) the ranker would interact with a length $K$ list $\mathcal{R}_t$ instead of a single arm; 2) *the user would generate click feedback under different click models that depend on examination probability.* Recall from the definition of click models, that in the position-based model the user would return at most $K$ clicks in one round, while in the cascade model, the user would return at most 1 click. Thus, if the attacker returns more than one click in the cascade model, its attack is unstealthy and inefficient.

## 4   Generalized List Poisoning Attack Strategy

In this section, we would propose a generalized list poisoning attack (`GA`) that misguides *any no-regret ranker* to place the target item at the first position of $\mathcal{R}_t$ for $T - o(T)$ times in expectation with $o(T)$ cost.

**`GA` against no-regret rankers.**   We briefly illustrate the process of `GA`. The strategy is summarized in Algorithm 1. The attacker first needs to design list $\mathcal{T}$, where $\mathcal{T} = \{\tilde{a}, \eta_1, ..., \eta_{K-1}\}$. Intuitively, items

$\eta_1, ..., \eta_{K-1}$ are utilized to fill in the rest of the positions when $\tilde{a}$ is in $\mathcal{R}_t$. The attacker would not manipulate item $\mathbf{a}_k^t \in \mathcal{T}$. If items $\mathbf{a}_k^t \in \tilde{\mathcal{D}}/\mathcal{T}$ are placed in the position $k$, the attacker would replace it with $\eta_{K+k-1}$ (lines 5-9 in Algorithm 1). This manipulation strategy can mislead the ranker to believe the attractiveness of items in $\tilde{D}/\mathcal{T}$ are smaller equal than $\alpha(\eta_K)$. Hence, list $\mathcal{T}$ would be deemed as the optimal list and $\tilde{a}$ is the item with the highest attractiveness. According to the definition of the no-regret, no-regret algorithms will present $\mathcal{T}$ for $T - o(T)$ times and the attacker can achieve its attack goal. Theorem 1 demonstrates that GA can efficiently attack any no-regret ranker.

**Theorem 1.** *GA can efficiently attack any no-regret ranker, i.e., $\mathbb{E}[N_T(\tilde{a})] \geq T - \tilde{R}(T)/\Delta_{\min}$ and $\mathcal{C}(T) \leq K\tilde{R}(T)/\Delta_{\min}$, where $\tilde{R}(T) = \mathbb{E}\Big[T\sum_{k=1}^K v(\mathcal{T}, \mathbf{a}_k^t, k) - \sum_{t=1}^T \Big(\mathbb{1}\big\{attack, \ \sum_{k=1}^K v(\tilde{\mathcal{R}}_t, \mathbf{a}_k^t, k)\big\} + \mathbb{1}\big\{do \ not \ attack, \ \sum_{k=1}^K v(\mathcal{R}_t, \mathbf{a}_k^t, k)\big\}\Big)\Big]$ is the regret upper bound of the victim algorithm, $\Delta_{\min} = \min_{\mathcal{R} \in \Pi_K(\Lambda)}(\sum_{k=1}^K v(\mathcal{T}, \mathbf{a}_k^t, k) - \sum_{k=1}^K v(\mathcal{R}, \mathbf{a}_k^t, k)) > 0$ and $\Lambda$ consists of $\mathcal{T}$ and $L + K - 1$ items with attractiveness smaller equals then $\alpha(\eta_K)$.*

**Remark 2.** *According to the Algorithm 1, $\tilde{R}(T)$ can represent the regret of the victim algorithm when it explores an environment that consists of an optimal list $\mathcal{T}$ and $L + K - 1$ items with attractiveness smaller equal than $\alpha(\eta_K)$. According to the definition of the no-regret, the victim algorithm will present $\mathcal{T}$ for $T - o(T)$ times and achieve a sublinear regret, which implies $\tilde{R}(T) = o(T)$. Accordingly, we have $\mathbb{E}[N_T(\tilde{a})] = T - o(T)$ and $\mathcal{C}(T) = o(T)$, which satisfies the definition of the efficient attack.*

**Remark 3** (Gap dependency)**.** *If the regret upper bound of the victim ranker satisfies $\tilde{R}(T) = O(L \log(T)/\Delta_{\min})$ for example Kveton et al. (2015a), then GA has $\mathbb{E}[N_T(\tilde{a})] = T - O(L^2 \log(T)/\Delta_{\min}^2)$ and $\mathcal{C}(T) = O(L^2 \log(T)/\Delta_{\min}^2)$. This is similar to the target arm triggered rate and attack cost of the action poisoning attack on MAB in Liu and Lai (2020b).*

**Remark 4.** *We note that Zoghi et al. (2017); Lattimore et al. (2018) suppose the position-based model satisfies $\chi(1) \geq \chi(2) \geq, ..., \chi(K)$. This is because they are primarily concerned with achieving sublinear regrets rather than the specific position of an item in list $\mathcal{R}_t$. Similarly, Zuo et al. (2023) also disregard the position of the target item in $\mathcal{R}_t$, as their setting assumes the $\tilde{a} \in \mathcal{R}_t$ is an efficient attack. We make this assumption on the fact that if $\chi(1) = \chi(2)$, both $(a_1, a_2, a_3, ..., a_K)$ and $(a_2, a_1, a_3, ..., a_K)$ can achieve zero regret in the position-based model. This contradicts the assumption of our stochastic click model, which states that only $\mathcal{R}^* = (a_1, a_2, a_3, ..., a_K)$ can achieve zero regret. More specifically, if $\chi(1) = \chi(2)$, then an algorithm can present $(a_2, a_1, a_3, ..., a_K)$ for $T - o(T)$ to achieve a $o(T)$ regret, and GA can not achieve its attack goal due to $\tilde{a}$ has the highest attractiveness and it will be placed on the second position. Besides, it's important to note that we can alleviate the assumption to $\chi(1) > \chi(2) \geq \cdots \geq \chi(K)$ and our GA can also achieve its attack goal in the position-based model. This is because we only focus on the position of the target item and disregard the positions of non-attractive items $\{\eta_k\}_{k=1}^{K-1}$. Furthermore, suppose $\chi(k) > \chi(k')$ is highly plausible because in real-world scenarios, it is impossible for this phenomenon to occur, as users inherently exhibit biases towards distinct positions.*

Although we do not include all rankers with $o(T)$ regret in the cascade model as no-regret rankers from Remark 1, we can still show that cascade ranking algorithms such as CascadeUCB1 (Kveton et al., 2015a) can be efficiently attacked by Algorithm 1.

**Theorem 2.** *Consider the ranker applies CascadeUCB1 and is attacked by Algorithm 1. The expected number of $\tilde{a}$ be placed at the first position till round $T$ satisfies $\mathbb{E}[N_T(\tilde{a})] = T - o(T)$ and the cost is $\mathcal{C}(T) = o(T)$.*

**Remark 5.** *We notice that most existing cascade rankers for OLTR problems are UCB-based algorithms (CascadeUCB1, CascadeKLUCB (Kveton et al., 2015a), and CascadeUCBV (Vial et al., 2022)). Hence, we believe GA can efficiently attack these algorithms and we can analyze the theoretical results similar to Theorem 2. We have also provided empirical results in our experiment section and Appendix that GA can efficiently attack CascadeUCB1, CascadeKLUCB, and the SOTA algorithm CascadeUCB-V.*

After the above observation, one may wonder why we don't utilize click poisoning strategy to achieve the same goal of GA, we here propose a motivated example.

**Example 2** (Limitation of click poisoning attack)**.** *Suppose the environment includes an item set $\mathcal{A} = \{a_1, a_2, a_3\}$ with attractiveness $\alpha(a_1) > \alpha(a_2) > \alpha(a_3)$ and $a_2$ is the target item. Besides, we suppose the*

*length of ranking list $K = 2$ and the attacker knows the preference between the items. The attacker reduces the click feedback of $a_1$ to 0 when $a_1 \in \mathcal{R}_t$, otherwise, does not attack. When attacking position-based OLTR algorithm, the effectiveness of the attack can be deemed as the attacker reducing the attractiveness of $a_1$ to 0 and holding the attractiveness of $a_2$ and $a_3$. Note that keeping the true attractiveness of target item $a_2$ unchanged is the key idea to achieve sublinear attack cost.*

*However, the above analysis for the position-based model cannot be applied to the Cascade model. Different from the PBM, the click probability of an item in the Cascade model is not only related to this item's attractiveness but also related to other items' attractiveness. Suppose item $a_1$ is placed at the first position of $\mathcal{R}_t$ and item $a_2$ is placed at the second position. If the user clicks $a_1$ and returns the click feedback (the click feedback of $a_1$ is 1 and the click feedback of $a_2$ is 0), this implies $a_2$ has not been examined by the user and the ranker does not update $a_2$'s attractiveness estimate. When the attacker reduces the click feedback of $a_1$ from 1 to 0, we can not simply deem the attacker reducing the attractiveness of $a_1$ to 0 and holding the attractiveness of $a_2$. This is because the clicks of two items in the Cascade model are 0s means the user examines **all** items and does not click either of them. Hence, the clicks will be interpreted by the victim algorithm as the attractiveness of $a_1$ and $a_2$ is 0, violating the idea that the attacker should keep true attractiveness of target item for sublinear cost.*

The above example shows that the simple click poisoning strategy cannot adapt to different click models, and its theoretical results need to be analyzed differently based on the click model's characteristics. However, GA perturbs the ranking list presented to the user and pretends the click feedback represents the user's interests in the original ranking list. The effectiveness of this strategy can be viewed as the attractiveness of items in the original list is replaced by the attractiveness of items in the perturbed list. This guarantees the click feedback always follows the **unknown click model**, making the attack easy to analyze and stealthy.

## 5 Attack-Then-Quit Strategy

In this section, we provide a click poisoning attack strategy that applied to rankers on general click models. We will demonstrate our strategy can *efficiently attack* some rankers under any click models. Previous reward poisoning attack strategies on multi-armed bandits only manipulate the reward of the non-target item and never perturb the reward of the target item (Jun et al., 2018; Liu and Shroff, 2019; Garcelon et al., 2020; Wang et al., 2022). However, the attack strategy proposed in this section perturbs the click feedback of every item. The attacker would spend a linear cost in a short period $T_1 = o(T)$ and then stop the attack and quit. Thus, we call our attack strategy the attack-then-quit (ATQ) strategy. As mentioned in the previous section, the BatchRank and TopRank can work in general stochastic click models, while other UCB-type algorithms can only work in specific click models (e.g., CascadeUCB1 or PBM-UCB). Hence, the BatchRank and TopRank can be deemed as the superior ranker to those UCB-type algorithms, which are worth more significant consideration.

### 5.1 Click Poisoning-Based ATQ

Algorithm 2 can efficiently attack elimination-based rankers (e.g., BatchRank) and divide-and-conquer-based rankers (e.g., TopRank). At the initialization stage, the attacker would receive target item $\tilde{a}$ and threshold $T_1$. In each round, if click feedback $\mathcal{C}_k^t$ is returned from $\tilde{a}$ and $\tilde{a} \in \mathcal{R}_t$, the attacker would set $\tilde{\mathcal{C}}_k^t = 1$, otherwise, $\tilde{\mathcal{C}}_k^t = 0$. Besides, as we mention in Assumption 1 and Definition 1, the item $a_k \notin \mathcal{R}_t$ would not be observed by the user in round $t$, and would not be clicked either. This implies their click feedback would be 0. In ATQ, the attacker does not (cannot) modify the click feedback of items $a_k \notin \mathcal{R}_t$ (line 9~10 in Algorithm 2). After scanning $\mathcal{C}_t$, the attacker would return $\tilde{\mathcal{C}}_t = (\tilde{\mathcal{C}}_1^t, ..., \tilde{\mathcal{C}}_L^t)$ to the ranker. Note that $\sum_{k=1}^{L} \tilde{\mathcal{C}}_k^t \leq 1$ (line 4-10 in Algorithm 2), hence Algorithm 2 can work stealthily and efficiently in different click models. Note that the divide-and-conquer and elimination nature ensures BatchRank and TopRank to quickly estimate the attractiveness of items and rank them accordingly. Our attack Algorithm 2 aims to mislead these rankers to 'believe' the target item is the best item with the smallest rounds. Once these rankers make this judgment, they would place the target item at the first position until $T$. In the following sections, we will demonstrate Algorithm 2 can efficiently attack BatchRank and TopRank with suitable $T_1$.

---

**Algorithm 2** Attack-Then-Quit (`ATQ`) Algorithm

---
1: **Inputs:** The target item $\tilde{a}$, threshold $T_1$
2: **for** $t = 1 : T_1$ **do**
3:     Observe $\mathcal{C}_t = (\mathcal{C}_1^t, ..., \mathcal{C}_L^t)$ and $\mathcal{R}_t = (\mathbf{a}_1^t, ..., \mathbf{a}_K^t)$
4:     **for** $k = 1 : L$ **do**
5:         **if** $a_k = \tilde{a}$ and $a_k \in \mathcal{R}_t$ **then**
6:             Set $\tilde{\mathcal{C}}_k^t = 1$
7:         **else if** $a_k \neq \tilde{a}$ and $a_k \in \mathcal{R}_t$ **then**
8:             Set $\tilde{\mathcal{C}}_k^t = 0$
9:         **else**
10:        Set $\tilde{\mathcal{C}}_k^t = \mathcal{C}_k^t$
11:    Return $\tilde{\mathcal{C}}_t = (\tilde{\mathcal{C}}_1^t, ..., \tilde{\mathcal{C}}_L^t)$ to the ranker

---

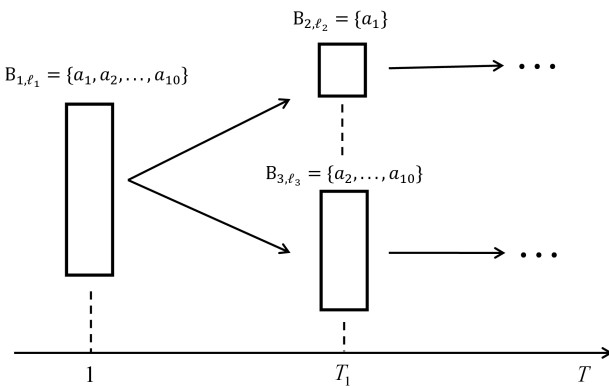

Figure 2: Process of the Algorithm 2 attacks BatchRank.

## 5.2 Attack on BatchRank

**Brief Explanation of BatchRank.** The BatchRank falls into the category of elimination-based algorithms (Zoghi et al., 2017). The BatchRank would begin with stage $\ell_1 = 0$ and the first batch $B_{1,\ell_1} = \mathcal{D}$. In stage $\ell_1$, every item would be explored for $n_{\ell_1} = 16\tilde{\Delta}_{\ell_1}^{-1}\log(T)$ times and $\tilde{\Delta}_{\ell_1}^{-1} = 2^{-\ell_1}$. Afterward, if BatchRank collects enough information to confirm the attractiveness of every item in a group is larger than the others, BatchRank then splits the initial batch into two sub-batches $B_{2,\ell_2}$ and $B_{3,\ell_3}$. Items that are considered with higher attractiveness (i.e., in sub-batch $B_{2,\ell_2}$) would be always placed before items with lower attractiveness (i.e., in sub-batch $B_{3,\ell_3}$). The BatchRank would restart with stage $\ell_2 = 0$ and $\ell_3 = 0$ and sub-batches $B_{2,\ell_2}$ and $B_{3,\ell_3}$. Batches would recursively split until round $T$. Intuitively, the action 'split' of BatchRank is similar to the elimination action in the elimination-based bandit algorithms (Even-Dar et al., 2006; Lykouris et al., 2018; Bogunovic et al., 2021). The details of BatchRank are provided in the Appendix.

Algorithm 2 can successfully attack BatchRank owing to BatchRank's elimination property. Algorithm 2 maximizes the returned clicks of the target item and minimizes the returned click of the non-target item in a short period $o(T)$. After this period, BatchRank regards the target item owning the highest attractiveness (i.e., split). Then the target item would be always placed at the first position of the list $\mathcal{R}_t$ and the attacker achieves the attack goal. We also provide Example 3 to further illustrate the process of Algorithm 2 attack BatchRank.

**Example 3.** *Consider the item set $\mathcal{D} = \{a_1, a_2, ..., a_{10}\}$ contains 10 items. The target item is $\tilde{a} = a_1$ with arbitrary attractiveness and item list $\mathcal{R}_t$ contains 3 positions. The BatchRank begins with $B_{1,\ell_1} = \mathcal{D}$ and recursively explores items. The attacker begins the attack at round 1 and ends at $T_1$. During $T_1$ rounds, the attacker returns $\tilde{\mathcal{C}}_k^t = 1$ if $\tilde{a} = a_k$ and $\tilde{a} \in \mathcal{R}_t$, otherwise $\tilde{\mathcal{C}}_k^t = 0$. This fraudulent strategy misleads BatchRank to believe the attractiveness of the non-target items $a_k \neq \tilde{a}$ is 0 and the attractiveness of $\tilde{a}$ is 1. After BatchRank finishes the split section, the batch $B_{1,\ell_1}$ split into $B_{2,\ell_2} = \{a_1\}$ and $B_{3,\ell_3} = \{a_2, ..., a_{10}\}$.*

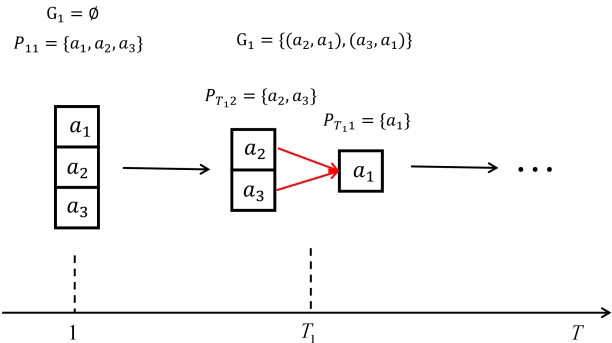

Figure 3: Process of the Algorithm 2 attacks TopRank.

*Sub-batch $B_{2,\ell_2}$ contains position 1 (i.e., $\mathbf{a}_1^t$) and $B_{3,\ell_3}$ contains position 2 and 3 (i.e., $\mathbf{a}_2^t$ and $\mathbf{a}_3^t$). The process is illustrated in Figure 2. The target item $\tilde{a}$ would always be placed at the first position of the $\mathcal{R}_t$ in the next $T - T_1$ rounds due to $B_{2,\ell_2}$ only contains $\tilde{a}$. The attack cost would be $o(T)$ because the attacker only manipulates the click feedback for $o(T)$ rounds.*

**Theorem 3** (Attack against BatchRank). *Consider the attacker aims to attack BatchRank with stochastic click models (Zoghi et al., 2017). If the attacker runs Algorithm 2 with $T_1 = 16L\log(T)$, we have $N_T(\tilde{a}) \geq T - T_1$ and the cost can be bounded by $\mathcal{C}(T) \leq KT_1$.*

Besides BatchRank, this attack idea can also be utilized to attack some rankers that do not belong to the elimination-based category, such as TopRank.

### 5.3 Attack on TopRank

**Brief Explanation of TopRank.** TopRank is a divide-and-conquer-based ranker (Lattimore et al., 2018). It begins with a blank graph $G_1 = \emptyset$. In round $t$, TopRank would establish blocks $\mathcal{P}_{t1}, ..., \mathcal{P}_{td}$ via graph $G_t$. The items in block $\mathcal{P}_{t1}$ would be placed at the first $|\mathcal{P}_{t1}|$ positions and the items in block $\mathcal{P}_{t2}$ would be placed at the next $|\mathcal{P}_{t2}|$ positions, and so on. During rounds 1 to $T$, TopRank would explore items with blocks, collect click information and compare attractiveness between items in the same block. If the collected evidence is enough to let TopRank regards the attractiveness of item $a_i$ as larger than the attractiveness of item $a_j$, a directional edge $(a_j, a_i)$ would be established. This behavior is similar to the 'split' action in BatchRank. Besides, graph $G_t$ would not contain cycles with high probability. If the graph contains at least one cycle, we consider TopRank would be out of control. Details of TopRank are provided in the appendix.

Note that if there exist edges from every non-target item to the target item and $G_t$ contains no cycle, then the target item would be isolated from the non-target items and would always be placed at the first position of $\mathcal{R}_t$. This is because the first block only contains the target item. We also provide Example 4 to specifically explain how Algorithm 2 attacks TopRank.

**Example 4.** *The process of the attack is shown in Figure 3. Consider the total item set $\mathcal{D} = \{a_1, a_2, a_3\}$ with 3 items. The length of the list $\mathcal{R}_t$ is $K = 2$ and the target item is $\tilde{a} = a_1$. The TopRank would start with block $\mathcal{P}_{11} = \mathcal{D}$ and $d = 1$ because the graph contains no edges at the beginning. In the first $T_1$ rounds, the attacker receives click feedback $\mathcal{C}_t$ and modifies click feedback $\tilde{\mathcal{C}}_k^t = 1$ if $a_k = \tilde{a}$ and $a_k \in \mathcal{R}_t$, otherwise $\tilde{\mathcal{C}}_k^t = 0$. After $T_1$, the edges $(a_k, \tilde{a})$, $k = 2, 3$ are established simultaneously. In the last $T - T_1$ rounds, the block $\mathcal{P}_{t1}$ would only contain $\tilde{a}$ and $\tilde{a}$ would always be placed at the first position of $\mathcal{R}_t$. Due to TopRank would only compare items' attractiveness in the same block, the edges from $\tilde{a}$ to $a_k \neq \tilde{a}$ would never be established and cycle would appear in $G_t$ with very low probability (will be explained in the proof of Theorem 4 in the appendix).*

**Theorem 4** (Attack against TopRank). *Consider the attacker aims to attack TopRank and TopRank runs in the click model in (Lattimore et al., 2018). If the attacker sets $T_1 = \frac{4\log(c/\delta)}{\frac{K}{L} + (1 - \sqrt{1 + 8K/L})/4}$ and runs Algorithm 2. The attacker can achieve $N_T(\tilde{a}) \geq T - T_1$ with probability at least $1 - (1/c + L^2)\delta$. The cost can be bounded by $\mathcal{C}(T) \leq KT_1$.*

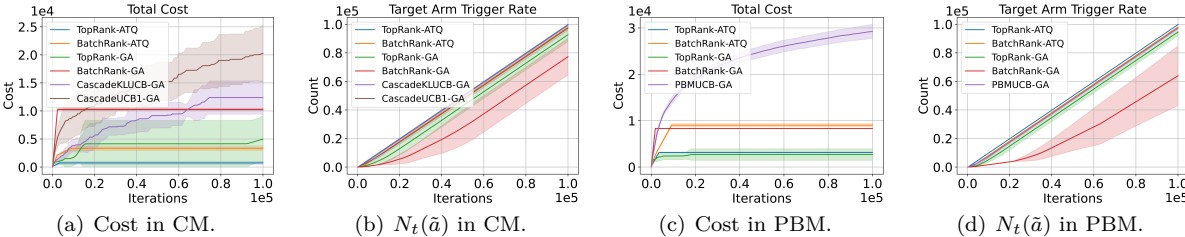

Figure 4: Synthetic data experiment: (a) total cost spend in the cascade model, (b) $N_t(\tilde{a})$ in the cascade model, (c) the total cost spend in the cascade model and (d) $N_t(\tilde{a})$ in the position-based model. We report averaged result and variance of 10 runs.

By choosing $\delta = 1/T$ and $c = 4\sqrt{2/\pi}/erf(\sqrt{2}) \approx 3.43$ which is same as in TopRank algorithm, we have $T_1 = O((L/K)\log T)$. Besides, readers should note that $1 - \delta L^2$ is the intrinsic probability of TopRank's $G_t$, $\forall t \in [T]$ contains cycles.

**Remark 6** (Why Theorem 3 and Theorem 4 is gap-independent?). *The reason the cost upper bound of the ATQ isn't gap-dependent is due to 1) the approach taken during the attack, and 2) the structure of the BatchRank and TopRank. The TopRank and BatchRank algorithms are classified as phase elimination algorithms. In the proofs of Theorems 3 and 4, we demonstrate that we can achieve the attack objective by maximize the click feedback of the target item and minimize the click feedback of the non-target items during the first elimination phase (see ATQ pseudo code). Through analysis, we determine that the length of the first elimination phase for BatchRank and TopRank is respectively bounded by $O\big(L\log(T)\big)$ and $O\big(\frac{L}{K}\log(T)\big)$ (this also implies that ATQ can compromise TopRank faster than BatchRank due to BatchRank's inferior performance compared to TopRank.), which is unrelated to the attractiveness gap. Besides, due to TopRank and BatchRank will randomly explore the whole item set in the first phase, we suppose the attack cost per round is $K$ due to the attacker should modify at most $K$ click feedback in each round. Accordingly, the finial cost upper bound do not depends to the attractiveness gaps.*

**Remark 7** (Comparison of ATQ performance with that of attack algorithms in the bandit domain). *Although we haven't provided an instance-dependent version of cost upper bound, we notice that when the victim algorithm is BatchRank and TopRank, the ATQ can achieve $\mathbb{E}[N_T(\tilde{a})] = T - O(L\log(T))$ with $\mathcal{C}(T) = O(L\log(T))$. This can match the performance of click poisoning attack on MAB (Theorem 1 and Theorem 2 of Jun et al. (2018)), i.e., $\mathbb{E}[N_T(\tilde{a})] = T - O(L\log(T))$ and $\mathcal{C}(T) = O(L\log(T))$.*

**Remark 8** (Why UCB-based algorithms can not be attacked by ATQ?). *The effect of ATQ relies on algorithms' phased elimination property. However, CascadeUCB and PBM-UCB belong to UCB-based algorithms. When the attacker stops, the true click feedback of other items will be revealed to the algorithms over time, leading UCB-based algorithms to realize the targeted item isn't the item with the highest attractiveness.*

## 6 Experiments

In the experiment section, we apply the proposed attack methods against the OLTR algorithms listed in Table 1 with their corresponding click models. We compare the effectiveness of our attack on synthetic data and real-world MovieLens dataset. For all our experiments, we use $L = 50$, $K = 5$ (the set up of Zoghi et al. (2017); Lattimore et al. (2018) is $L = 10$ and $K = 5$) and $T = 10^5$. For ATQ, we set the $T_1$ in Algorithm 2 by Theorem 3 and Theorem 4.

### 6.1 Synthetic Data

First, we verify the effectiveness of our proposed attack strategies on synthetic data. We generate a size-$L$ item set $\mathcal{D}$, in which each item $a_k$ is related to a unique attractiveness score $\alpha(a_k)$. Each attractiveness score $\alpha(a_k)$ is drawn from a uniform distribution $U(0, 1)$. We randomly select a suboptimal target item $\tilde{a}$. Figure 4 shows the results and variances of 10 runs.

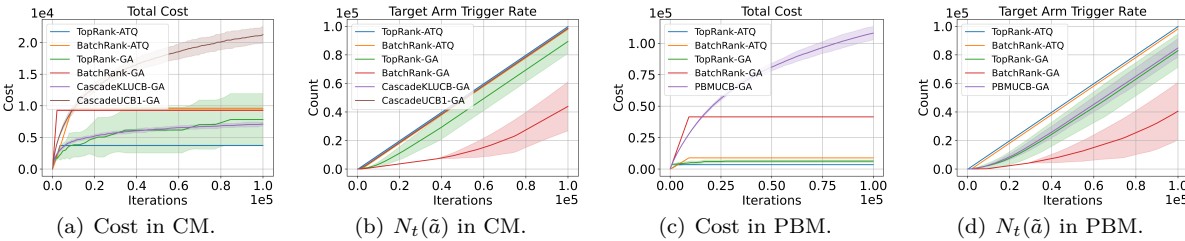

(a) Cost in CM.     (b) $N_t(\tilde{a})$ in CM.     (c) Cost in PBM.     (d) $N_t(\tilde{a})$ in PBM.

Figure 5: MovieLens experiment: (a) the total cost spend in the cascade model, (b) $N_t(\tilde{a})$ in the cascade model, (c) the total cost spend in the cascade model and (d) $N_t(\tilde{a})$ in the position-based model. We report averaged result and variance of 10 runs.

Table 1: Target ranking algorithms and their applied click models

| Algorithm | Click model |
|---|---|
| BatchRank (Zoghi et al., 2017) | Stochastic click model |
| TopRank (Lattimore et al., 2018) | Stochastic click model |
| PBM-UCB (Lagrée and Vernade, 2016) | Position-based model |
| CascadeUCB1 (Kveton et al., 2015a) | Cascade model |
| CascadeKLUCB (Kveton et al., 2015a) | Cascade model |

In Figures 4(a) and 4(b), we plot the results of the `GA` against CascadeUCB1, CascadeKLUCB, BatchRank, and TopRank, and the `ATQ` against BatchRank and TopRank in the cascade model. Both attack strategies can efficiently misguide the rankers to place the target item at the first position for $T - o(T)$ times as shown in Figure 4(b), and the cost of the attack is sublinear as shown in Figure 4(a). The `GA` is cost-efficient when attacking all four algorithms. We can observe that when it attacks TopRank and BatchRank, the cost would not increase after some periods (similar to the `ATQ`'s results). This is when the TopRank and BatchRank believe the target item and the auxiliary items have a relatively higher attractiveness than the other items, they would only put the target item and the auxiliary items in $\mathcal{R}_t$. Besides, when attacking TopRank and BatchRank, the growth rate of `GA`'s target arm pulls $N_t(\tilde{a})$ slowly increased from 0.2 per iteration to 1 per iteration. This is because the `GA` does not manipulate the items in $\mathcal{T}$ and the TopRank and BatchRank need time to confirm the target item has a higher attractiveness than $\{\eta_k\}_{k=1}^{K-1}$. Hence, the smaller the gap between $\tilde{a}$ and $\eta_1$, the larger the confirmed time. Compare with the `GA`, the `ATQ` can also efficiently attack BatchRank and TopRank with a sublinear cost. However, its $N_T(\tilde{a})$ is almost $T$, which is relatively larger than `GA`'s $N_T(\tilde{a})$. This is because the `ATQ` is specifically designed for divide-and-conquer-based algorithms like TopRank and BatchRank. The `ATQ` can maximize the target item's click number and misguide these algorithms to believe the target item is the best in the shortest period.

Figures 4(c) and 4(d) report the results in the position-based model. We can observe that the spending cost of the `GA` on the PBM-UCB is slightly larger than the spending cost on the CascadeKLUCB and CascadeUCB1. Besides, although the `GA` can let the TopRank believe the target item is the best item in almost 500 iterations, it still needs a large number of iterations (around $6 \times 10^4$ iterations) to make the BatchRank make such a decision. From the results of the two models, the `ATQ` is obviously more effective than the `GA` when the target algorithms are TopRank and BatchRank.

## 6.2 Experiments on Real-World Data

We also evaluate the proposed attacks on MovieLens dataset (Harper and Konstan, 2016). We first split the dataset into train and test data subsets. Using the training data, we compute a $d$-rank SVD approximation, which is used to compute a mapping from movie rating to the probability that a user selected at random would rate the movie with 3 stars or above. We use the learned probability to simulate user's clicks given the ranking list. We refer the reader to the Appendix C of (Vial et al., 2022) for further details. Figure 5 shows the attack results of our attack strategy averaged over 10 rounds.

We can observe that the trends in Figure 5 are similar to those in Figure 4, and the two attack algorithms are again able to efficiently fool the OLTR algorithms. In the cascade model, we see that successfully attacking CascadeKLUCB, TopRank, and BatchRank with `GA` only needs a relatively low cost, and the cost is higher when the target is CascadeUCB1. Besides, the `ATQ` strategy can still outperform the `GA` in $N_T(\tilde{a})$ when the target algorithms are TopRank and BatchRank. In the position-based model, the results are similar to the results in the cascade model, and the cost spent in the PBM-UCB is larger than the cost spent in the other algorithms.

## 7  Conclusion

In this paper, we study adversarial attacks on online learning to rank. Different from the poisoning attacks studied in the multi-armed bandits setting where reward or action is manipulated, the attacker manipulates binary click feedback instead of reward and item list instead of a single action in our model. In addition, due to the interference of the click models, it is difficult for the attacker to precisely control the ranker behavior under different unknown click models with simple click manipulation. Based on this insight, we developed the `GA` that can efficiently attack any no-regret ranking algorithm. Moreover, we also proposed the `ATQ` that follows the click poisoning idea, which can efficiently attack BatchRank and TopRank. Finally, we presented experimental results based on synthetic data and real-world data that validated the cost-efficient and effectiveness of our attack strategies.

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

# A  Notations

For clarity, we collect the frequently used notations in this paper.

| | |
|---|---|
| $\mathcal{D}$ | Total item set |
| $\mathcal{R}_t$ | $K$-length item list be shown to the user in round $t$ |
| $\mathcal{R}^*$ | Optimal list |
| $\tilde{\mathcal{R}}_t$ | Manipulated list in round $t$ |
| $\mathcal{C}_t$ | Click feedback list in round $t$ |
| $\tilde{\mathcal{C}}_t$ | Manipulated click feedback list in round $t$ |
| $\mathcal{T}$ | Ordered list $(\tilde{a}, \bar{a}_1, ..., \bar{a}_K)$ |
| $\tilde{a}$ | Target item |
| $a_k$ | $k$-th most attractive item in $\mathcal{D}$ |
| $\bar{a}_k$ | $k$-th most attractive auxiliary item |
| $\eta_k$ | Particular item in the list poisoning attack |
| $\alpha(a_k)$ | Attractiveness of item $a_k$ |
| $\mathbf{a}_k^t$ | Item on the $k$-th position in $\mathcal{R}_t$ |
| $\tilde{\mathbf{a}}_k^t$ | Manipulated item on the $k$-th position in $\tilde{\mathcal{R}}_t$ |
| $\mathcal{C}_k^t$ | Click feedback of item $a_k$ in round $t$ |
| $\tilde{\mathcal{C}}_k^t$ | Manipulated click feedback of the item $a_k$ in round $t$ |
| $v(\mathcal{R}_t, \mathbf{a}_k^t, k)$ | Click probability of item at the $k$-th position in round $t$ |
| $R(T)$ | Cumulative regret in $T$ rounds |
| $\mathcal{C}(T)$ | Total cost in $T$ rounds |
| $N_t(a_k)$ | Number of item $a_k$ be placed at the first position in $t$ rounds |
| $\mathcal{N}_t(a_k)$ | Number of item $a_k$ be examined in $t$ rounds |
| $T$ | Total number of interaction |
| $T_1$ | Input threshold value of the attack-then-quit algorithm |
| **BatchRank** | |
| $b$ | Batch index |
| $\ell$ | Stage index |
| $B_{b,\ell}$ | $b$-th batch explored in stage $\ell$ |
| $\boldsymbol{n}_\ell$ | Exploration number of item in batch $B_{b,\ell}$ in stage $\ell$ |
| $\mathcal{C}_{b,\ell}(a_k)$ | Total received click number of item $a_k$ during stage $\ell$ |
| $\hat{\mathcal{C}}_{b,\ell}(a_k)$ | Attractiveness estimator of item $a_k$ in stage $\ell$ |
| $U_{b,\ell}(a_k)$ | Upper confidence bound of item $a_k$ in stage $\ell$ |
| $L_{b,\ell}(a_k)$ | Lower confidence bound of item $a_k$ in stage $\ell$ |
| **TopRank** | |
| $G_t$ | Auxiliary graph in round $t$ |
| $(a_j, a_i)$ | Directional edge from item $a_j$ to item $a_i$ |
| $\mathcal{P}_{tc}$ | $c$-th block in round $t$ |
| $S_{tij}$ | Sum of the $U_{tij}$ from round 1 to $t$ |
| $N_{tij}$ | Sum of the absolute value of $U_{tij}$ from round 1 to $t$ |

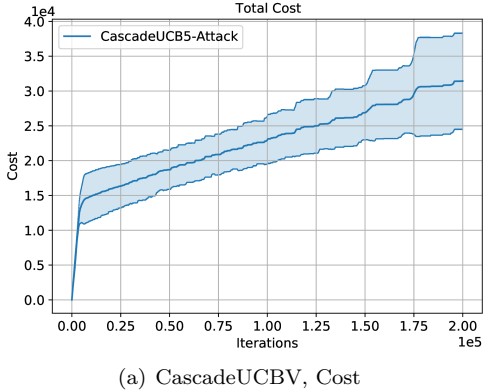

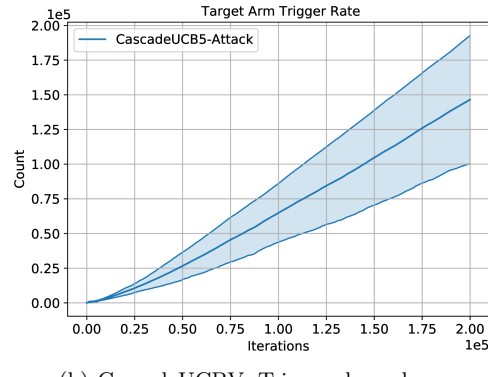

(a) CascadeUCBV, Cost

(b) CascadeUCBV, Triggered number

Figure 6: Experimental results of `GA` attacks CascadeUCBV. We report the average results and variance of 10 rounds.

## B  Additional Experiments

In the main text, we have shown that `GA` can efficiently attack CascadeUCB1 and CascadeKLUCB. In this subsection, we show that `GA` can also efficiently attack the SOTA cascade ranker CascadeUCBV (Vial et al., 2022). The attack results are provided in Figure 6. Apparently, the `GA` can successfully reach the attack goal. Based on these results, we believe `GA` can efficiently attack almost every existing UCB-type cascade ranker.

## C  Proof of Theorem 1

Recall that when the attacker implements Algorithm 1, the optimal list becomes $\mathcal{R}^* = \mathcal{T} = (\tilde{a}, \eta_1, ..., \eta_{K-1})$ due to the attractiveness of items belong to $\tilde{\mathcal{D}}\backslash\mathcal{T}$ is smaller than $\alpha(\eta_{K-1})$ (i.e., $\alpha(\eta_K) < \alpha(\eta_{K-1}) \leq \alpha(\tilde{a})$). We should also notice that the target item $\tilde{a}$ has the highest attractiveness. Now recall

$$\mathbb{E}[N_T(\tilde{a})] = \mathbb{E}\left[\sum_{t=1}^{T} \mathbb{1}\{\mathbf{a}_1^t = \tilde{a}\}\right] = T - \mathbb{E}\left[\sum_{t=1}^{T} \mathbb{1}\{\mathbf{a}_1^t \neq \tilde{a}\}\right], \tag{7}$$

then based on Definition 1 and Assumption 1, we know that if $\mathbf{a}_1^t \neq \tilde{a}$, the ranker would suffer from regret in round $t$. Due to the per step regret is at least $\Delta_{\min} = \min_{\mathcal{R}\in\Pi_K(\Lambda)}(\sum_{k=1}^{K} v(\mathcal{T}, \mathbf{a}_k^t, k) - \sum_{k=1}^{K} v(\mathcal{R}, \mathbf{a}_k^t, k)) > 0$, where $\Lambda$ consists of $\mathcal{T}$ and $L + K - 1$ items with attractiveness smaller equals then $\alpha(\eta_K)$. Then based on the definition of regret, we can finally derive

$$\mathbb{E}[N_T(\tilde{a})] \leq T - \frac{\tilde{R}(T)}{\Delta_{\min}}, \tag{8}$$

where $\tilde{R}(T) = \mathbb{E}\left[T\sum_{k=1}^{K} v(\mathcal{T}, \mathbf{a}_k^t, k) - \sum_{t=1}^{T}\left(\mathbb{1}\left\{attack, \sum_{k=1}^{K} v(\tilde{\mathcal{R}}_t, \mathbf{a}_k^t, k)\right\} + \mathbb{1}\left\{do\ not\ attack, \sum_{k=1}^{K} v(\mathcal{R}_t, \mathbf{a}_k^t, k)\right\}\right)\right]$ is the regret upper bound of the victim ranker.

Besides, according to the line 4 of the Algorithm 1, the cost of Algorithm 1 can be bounded by

$$\mathcal{C}(T) \leq \mathbb{E}\left[\sum_{t=1}^{T}\sum_{k=1}^{K} \mathbb{1}\{\mathbf{a}_k^t \in \tilde{\mathcal{D}}\backslash\mathcal{T}\}\right] \leq K\mathbb{E}\left[\sum_{t=1}^{T} \mathbb{1}\{\mathcal{R}_t\backslash\mathcal{T} \neq \emptyset\}\right]. \tag{9}$$

Due to the optimal list becomes $\mathcal{T}$ during the attack, if $\mathcal{R}_t\backslash\mathcal{T} \neq \emptyset$, the per step regret is at least $\Delta_{\min}$. Thus, the cost can be bounded by

$$\mathcal{C}(T) \leq K\mathbb{E}\left[\sum_{t=1}^{T} \mathbb{1}\{\mathcal{R}_t\backslash\mathcal{T} \neq \emptyset\}\right] \leq \frac{K\tilde{R}(T)}{\Delta_{\min}}. \tag{10}$$

Therefore, if the target ranker can present the best list $\mathcal{T}$ for $T - o(T)$ times in its click model under Assumption 1 (the definition of the no-regret ranker), then $\tilde{R}(T) = o(T)$ (GA would not attack the victim ranker when it presents $\mathcal{T}$), $\mathbb{E}[N_T(\tilde{a})] = T - o(T)$ and the cost of Algorithm 1 would be sublinear. According to Definition 3 and our deduction, we can conclude that if a ranker belongs to no-regret rankers, it can be efficiently attacked by Algorithm 1. Here finish the proof of Theorem 1.

# D   Proof of Theorem 2

## D.1   Introduction of CascadeUCB1

The pseudo-code of the CascadeUCB1 is provided as follows.

---

**Algorithm 3** CascadeUCB1 (Kveton et al., 2015a)

---
1: **Input:** Item set $\mathcal{D}$
2: **for** $k = 1 : L$ **do**
3:     Explore item $a_k$ and derive $\mathcal{C}_k^0$
4:     Set $\mathcal{N}_0(a_k) = 1$ and $\hat{\alpha}_1(a_k) = \mathcal{C}_k^0$
5: **for** $t = 1 : T$ **do**
6:     **for** $k = 1 : L$ **do**
7:         Compute $UCB_t(a_k)$
8:     Let $\mathbf{a}_1^t, ..., \mathbf{a}_K^t$ be $K$ items with largest UCBs and set $\mathcal{R}_t = (\mathbf{a}_1^t, ..., \mathbf{a}_K^t)$
9:     Observe click feedback $\mathcal{C}_t$
10:    **for** $k = 1 : K$ **do**
11:        **if** $\mathbf{a}_k^t$ is clicked **then**
12:            Set $s = k$
13:    **for** $k = 1 : L$ **do**
14:        Set $\mathcal{N}_t(a_k) = \mathcal{N}_{t-1}(a_k)$
15:    **for** $k = 1 : s$ **do**
16:        Set $\mathcal{N}_t(\mathbf{a}_k^t) = \mathcal{N}_t(\mathbf{a}_k^t) + 1$
17:        $\hat{\alpha}_{\mathcal{N}_t(\mathbf{a}_k^t)}(\mathbf{a}_k^t) = \frac{\mathcal{N}_{t-1}(\mathbf{a}_k^t)\hat{\alpha}_{\mathcal{N}_{t-1}(\mathbf{a}_k^t)}(\mathbf{a}_k^t) + \mathbb{1}\{s=k\}}{\mathcal{N}_t(\mathbf{a}_k^t)}$

---

We let $\mathcal{N}_t(a_k)$ denotes the number of item $a_k$ be examined till round $t$. The upper confidence bound is defined as $UCB_t(a_k) = \hat{\alpha}_{\mathcal{N}_{t-1}(a_k)}(a_k) + 3\sqrt{(\log(t-1))/\mathcal{N}_{t-1}(a_k)}$.

## D.2   Proof of Theorem 2

The proof of Theorem 2 relies on the following lemmas.

**Lemma 1** (The Hoeffding inequality)**.** *Let $X_1, X_2, ..., X_n$ i.i.d drawn from a Bernoulli distribution, $\bar{X} = \frac{1}{n}\sum_{i=1}^n X_i$ and $\mathbb{E}[X]$ be the mean, then*

$$P(\bar{X} - \mathbb{E}[X] \leq -a) \leq e^{-na^2/2}. \tag{11}$$

**Lemma 2.** *Consider item $a_1$ is the item with the highest attractiveness and $a_k \neq a_1$. When the principal runs the CascadeUCB1, the expected number of $a_k$ be placed at the first position till round $T$ can be bounded by $\mathbb{E}[N_T(a_k)] \leq 3 + 81\log(T)/\Delta_k^2$, where $\Delta_k = \alpha(a_1) - \alpha(a_k)$.*

*Proof of Lemma 2.* We first decompose $\mathbb{E}[N_T(a_k)]$ as follows

$$\mathbb{E}[N_T(a_k)] \leq 1 + \mathbb{E}\left[\sum_{t=1}^{T} \mathbb{1}\left\{\mathbf{a}_1^t = a_k, \ N_{t-1}(a_k) < \frac{81 \log(T)}{\Delta_k^2}\right\}\right] + \mathbb{E}\left[\sum_{t=1}^{T} \mathbb{1}\left\{\mathbf{a}_1^t = a_k, \ N_{t-1}(a_k) \geq \frac{81 \log(T)}{\Delta_k^2}\right\}\right]$$

$$\leq 1 + \frac{81 \log(T)}{\Delta_k^2} + \mathbb{E}\left[\sum_{t=1}^{T} \mathbb{1}\left\{\mathbf{a}_1^t = a_k, \ N_{t-1}(a_k) \geq \frac{81 \log(T)}{\Delta_k^2}\right\}\right]$$

$$\leq 1 + \frac{81 \log(T)}{\Delta_k^2} + \sum_{t=1}^{T} P\left(UCB_t(a_k) \geq UCB_t(a_1), \ N_{t-1}(a_k) \geq \frac{81 \log(T)}{\Delta_k^2}\right).$$

$$(12)$$

By union bound, we then decompose and bound probability $P\left(UCB_t(a_k) \geq UCB_t(a_1), \ N_{t-1}(a_k) \geq 81 \log(T)/\Delta_k^2\right)$

$$P\left(UCB_t(a_k) \geq UCB_t(a_1), \ N_{t-1}(a_k) \geq \frac{81 \log(T)}{\Delta_k^2}\right)$$

$$\leq \sum_{\lambda=1}^{t-1} \sum_{\sigma \geq \frac{81 \log(T)}{\Delta_k^2}}^{t-1} P\left(UCB_t(a_k) \geq UCB_t(a_1) \middle| \mathcal{N}_{t-1}(a_k) = \sigma, \ \mathcal{N}_{t-1}(a_1) = \lambda\right).$$

$$(13)$$

The inequality holds due to $\mathcal{N}_{t-1}(a_k) \geq N_{t-1}(a_k)$. We further upper bound $P\left(UCB_t(a_k) \geq UCB_t(a_1) \middle| \mathcal{N}_{t-1}(a_k) = \sigma, \ \mathcal{N}_{t-1}(a_1) = \lambda\right)$. Consider for $1 \leq \lambda \leq t-1$ and $81 \log(T)/\Delta_k^2 \leq \sigma \leq t-1$, we have

$$P\left(UCB_t(a_k) \geq UCB_t(a_1) \middle| \mathcal{N}_{t-1}(a_k) = \sigma, \ \mathcal{N}_{t-1}(a_1) = \lambda\right)$$

$$\leq P\left(\hat{\alpha}_{\mathcal{N}_{t-1}(a_k)}(a_k) + 3\sqrt{\frac{\log(T)}{\mathcal{N}_{t-1}(a_k)}} + \frac{\Delta_k}{3} \geq \hat{\alpha}_{\mathcal{N}_{t-1}(a_1)}(a_1) + 3\sqrt{\frac{\log(T)}{\mathcal{N}_{t-1}(a_1)}} \middle| \mathcal{N}_{t-1}(a_k) = \sigma, \ \mathcal{N}_{t-1}(a_1) = \lambda\right)$$

$$\leq P\left(\hat{\alpha}_{\mathcal{N}_{t-1}(a_k)}(a_k) + \frac{2\Delta_k}{3} \geq \hat{\alpha}_{\mathcal{N}_{t-1}(a_1)}(a_1) + 3\sqrt{\frac{\log(T)}{\mathcal{N}_{t-1}(a_1)}} \middle| \mathcal{N}_{t-1}(a_k) = \sigma, \ \mathcal{N}_{t-1}(a_1) = \lambda\right)$$

$$\leq P\left(\hat{\alpha}_{\mathcal{N}_{t-1}(a_k)}(a_k) + \alpha(a_1) - \alpha(a_k) \geq \frac{\Delta_k}{3} + \hat{\alpha}_{\mathcal{N}_{t-1}(a_1)}(a_1) + 3\sqrt{\frac{\log(T)}{\mathcal{N}_{t-1}(a_1)}} \middle| \mathcal{N}_{t-1}(a_k) = \sigma, \ \mathcal{N}_{t-1}(a_1) = \lambda\right).$$

$$(14)$$

The first inequality relies on the definition of the $UCB_t(a_k)$. The second inequality holds because $\sigma \geq 81 \log(T)/\Delta_k^2$. The third inequality holds because $\Delta_k = \alpha(a_1) - \alpha(a_k)$.

Based on the Hoeffding inequality, we have for any $\lambda \geq 1$ and $\sigma \geq 81 \log(T)/\Delta_k^2$

$$P\left(\alpha(a_1) - \hat{\alpha}_{\mathcal{N}_{t-1}(a_1)}(a_1) \geq 3\sqrt{\frac{\log(T)}{\mathcal{N}_{t-1}(a_1)}} \middle| \mathcal{N}_{t-1}(a_1) = \lambda\right) \leq \frac{1}{T^{9/2}}$$

$$P\left(\hat{\alpha}_{\mathcal{N}_{t-1}(a_k)}(a_k) - \alpha(a_k) \geq \frac{\Delta_k}{3} \middle| \mathcal{N}_{t-1}(a_k) = \sigma\right) \leq \frac{1}{T^{9/2}}.$$

$$(15)$$

The last term of (14) can be further bounded by

$$
P\left(\hat{\alpha}_{\mathcal{N}_{t-1}(a_k)}(a_k) + \alpha(a_1) - \alpha(a_k) \geq \frac{\Delta_k}{3} + \hat{\alpha}_{\mathcal{N}_{t-1}(a_1)}(a_1) + 3\sqrt{\frac{\log(T)}{\mathcal{N}_{t-1}(a_1)}} \bigg| \mathcal{N}_{t-1}(a_k) = \sigma, \ \mathcal{N}_{t-1}(a_1) = \lambda\right)
$$

$$
\leq P\left(\hat{\alpha}_{\mathcal{N}_{t-1}(a_k)}(a_k) - \alpha(a_k) \geq \frac{\Delta_k}{3} \bigg| \mathcal{N}_{t-1}(a_k) = \sigma\right) + P\left(\alpha(a_1) - \hat{\alpha}_{\mathcal{N}_{t-1}(a_1)}(a_1) \geq 3\sqrt{\frac{\log(T)}{\mathcal{N}_{t-1}(a_1)}} \bigg| \mathcal{N}_{t-1}(a_1) = \lambda\right)
$$

$$
\leq \frac{1}{T^{9/2}} + \frac{1}{T^{9/2}} \leq \frac{2}{T^{9/2}}.
$$

(16)

The first inequality holds due to the union bound and the last inequality holds due to (15).

With the fact that

$$
\sum_{t=1}^{T} \sum_{\lambda=1}^{t-1} \sum_{\sigma \geq \frac{81\log(T)}{\Delta_k^2}}^{t-1} \frac{2}{T^{9/2}} \leq \frac{2}{T^{3/2}} \leq 2.
$$

(17)

In the light of (17), the total expected number of $a_k$ been placed at the first position can be bounded by

$$
\mathbb{E}[N_T(a_k)] \leq 3 + \frac{81\log(T)}{\Delta_k^2}.
$$

(18)

Here finish the proof of Lemma 2. $\qquad\square$

*Proof of Theorem 2.* With Lemma 2, we can bound the total expected number of $a_k \neq a_1$ being placed at the first position till round $T$. Thus, from round 1 to round $T$, the expected number of CascadeUCB1 place item $a_1$ at the first position satisfies

$$
N_T(a_1) \geq T - \sum_{k=2}^{L}\left(3 + \frac{81\log(T)}{\Delta_k^2}\right).
$$

(19)

Remember when the attacker implements attack Algorithm 1, the target item would become the item with the highest attractiveness. The rest of the items consist of $\{\eta_k\}_{k=1}^{K-1}$ and $L + K - 1$ items with attractiveness at most $\alpha(\eta_K)$. Therefore, when Algorithm 1 attacks the CascadeUCB1, $N_T(\tilde{a})$ can be lower bounded by

$$
\mathbb{E}[N_T(\tilde{a})] \geq T - \sum_{k=1}^{K-1}\left(\frac{3 + 81\log(T)}{(\alpha(\tilde{a}) - \alpha(\eta_k))^2}\right) - \sum_{s=1}^{L+K-1}\left(\frac{3 + 81\log(T)}{(\alpha(\tilde{a}) - \alpha(\eta_K))^2}\right).
$$

(20)

Besides, according to the line 4 of Algorithm 1, the cost of Algorithm 1 attack CascadeUCB1 can be bounded by

$$
\mathcal{C}(T) \leq K\mathbb{E}\left[\sum_{t=1}^{T} \mathbb{1}\{\mathcal{R}_t \backslash \mathcal{T} \neq \emptyset\}\right].
$$

(21)

It is worth noting that Algorithm 1 only manipulates items in list $\mathcal{R}_t$, hence the cost generates in one round is at most $K$. Recall the definition of regret in the cascade model

$$
R(T) = \mathbb{E}\left[T\sum_{k=1}^{K} v(\mathcal{T}, \mathbf{a}_k^t, k) - \sum_{t=1}^{T}\sum_{k=1}^{K} v(\mathcal{R}_t, \mathbf{a}_k^t, k)\right]
$$

$$
= \mathbb{E}\left[T\left(1 - (1 - \alpha(\tilde{a}))\prod_{k=1}^{K-1}(1 - \alpha(\bar{a}_k))\right) - \sum_{t=1}^{T}\left(1 - \prod_{k=1}^{K}(1 - \alpha(\mathbf{a}_k^t))\right)\right].
$$

(22)

The total regret is generated by $K$ positions. Algorithm 1 only attacks when $\mathcal{R}_t \backslash \mathcal{T} \neq \emptyset$. And situation $\mathcal{R}_t \backslash \mathcal{T} \neq \emptyset$ implies there is at least one item $\notin \mathcal{T}$ be placed in the $\mathcal{R}_t$ and its attractiveness is reduced to at most $\alpha(\eta_K)$. Due to when $\mathcal{R}_t \backslash \mathcal{T} \neq \emptyset$, the number of items is placed in $\mathcal{R}_t$ and belongs to $\tilde{\mathcal{D}} \backslash \mathcal{T}$ is at least 1. Then for the cascade model, the regret generates in round $t$ is at least

$$
\sum_{k=1}^{K} \left( v(\mathcal{T}, \mathbf{a}_k^t, k) - v(\mathcal{R}_t, \mathbf{a}_k^t, k) \right)
$$

$$
\geq 1 - (1 - \alpha(\tilde{a})) \prod_{k=1}^{K-1} (1 - \alpha(\eta_k)) - 1 + (1 - \alpha(\tilde{a}))(1 - \alpha(\eta_K)) \prod_{k=1}^{K-2} (1 - \alpha(\eta_k)) \tag{23}
$$

$$
= (\alpha(\eta_1) - \alpha(\eta_K))(1 - \alpha(\tilde{a})) \prod_{k=1}^{K-2} (1 - \alpha(\eta_k)).
$$

The first inequality holds due to $\alpha(\eta_{K-1})$ has the lowest attractiveness in $\mathcal{T}$. With the above derivation, we can derive when $\mathcal{R}_t \backslash \mathcal{T} \neq \emptyset$, the regret generates in each round is at least $(\alpha(\eta_1) - \alpha(\eta_K))(1 - \alpha(\tilde{a})) \prod_{k=1}^{K-2}(1 - \alpha(\eta_k))$. With this in mind, we can further bound the total cost by

$$
\mathcal{C}(T) \leq K \mathbb{E} \left[ \sum_{t=1}^{T} \mathbb{1}\{\mathcal{R}_t \backslash \mathcal{T} \neq \emptyset\} \right] \leq \frac{KR(T)}{(\alpha(\eta_1) - \alpha(\eta_K))(1 - \alpha(\tilde{a})) \prod_{k=1}^{K-2}(1 - \alpha(\eta_k))} \tag{24}
$$

Due to the regret of the CascadeUCB1 satisfies $R(T) = o(T)$, the cost of Algorithm 1 would be sublinear. We conclude that the CascadeUCB1 can be efficiently attacked by Algorithm 1. Here finish the proof of Theorem 2. $\qquad\square$

## E  Proof of Theorem 3

### E.1  Introduction of BatchRank

We here specifically illustrate details of BatchRank. The pseudo-code of the BatchRank is provided as follows.

---
**Algorithm 4** BatchRank (Zoghi et al., 2017)

---
1: **Initialize:** $b_{\max} = 1$, $\mathbf{I}_1 = (\mathbf{I}_1(1) = 1, \mathbf{I}_1(2) = K)$, $\ell_1 = 0$, $B_{1,0} = \mathcal{D}$, $\mathbb{B} = \{1\}$
2: **for** $b = 1 : K$ **do**
3:     **for** $\ell = 0 : T - 1$ **do**
4:         **for all** $a_k \in \mathcal{D}$ **do**
5:             $\mathcal{C}_{b,\ell}(a_k) = 0$, $\boldsymbol{n}_{b,\ell}(a_k) = 0$
6: **for** $t = 1 : T$ **do**
7:     **for all** $b \in \mathbb{B}$ **do**
8:         DisplayBatch($t,b$)
9:     **for all** $b \in \mathbb{B}$ **do**
10:        CollectClicks($t,b$)
11:     **for all** $b \in \mathbb{B}$ **do**
12:        UpdateBatch($t,b$)

---

The BatchRank explores items with batches, which are indexed by $b$. The BatchRank would begin with stage $\ell_1 = 0$, batch index $b = 1$, and the first batch $B_{b,\ell_1} = \mathcal{D}$. The first position in batch $b$ is indexed by $\mathbf{I}_b(1)$ and the last position is indexed by $\mathbf{I}_b(2)$, and the number of positions in batch $b$ is $\text{len}(b) = \mathbf{I}_b(1) - \mathbf{I}_b(2) + 1$. The first batch $B_{b,\ell_1}$ contains all the positions in $\mathcal{R}_t$. In stage $\ell_1$, every item in $B_{b,\ell_1}$ would be explored for $\boldsymbol{n}_{\ell_1} = 16\tilde{\Delta}_{\ell_1}^{-2} \log(T)$ times (DisplayBatch) and $\tilde{\Delta}_{\ell_1}^{-1} = 2^{-\ell_1}$. Afterward, the BatchRank would estimate the attractiveness of item $a_k$ as

$$
\hat{\mathcal{C}}_{b,\ell}(a_k) = \mathcal{C}_{b,\ell}(a_k)/\boldsymbol{n}_\ell. \tag{25}
$$

---

**Algorithm 5** DisplayBatch

---

1: **Input:** batch index $b$, time $t$
2: Set $\ell = \ell_b$
3: Let $a_1, ..., a_{|B_{b,\ell}|}$ be a random permutation of items in $B_{b,\ell}$ such that $\boldsymbol{n}_{b,\ell}(a_1) \leq ... \leq \boldsymbol{n}_{b,\ell}(a_{|B_{b,\ell}|})$
4: Let $\pi \in \prod_{len(b)}([len(b)])$ be a random permutation of position assignments
5: **for** $k = \mathbf{I}_b(1) : \mathbf{I}_b(2)$ **do**
6: $\quad \mathbf{a}_k^t = a_{\pi(k-\mathbf{I}_b(1)+1)}$

---

---

**Algorithm 6** CollectClicks

---

1: **Input:** batch index $b$, time $t$
2: Set $\ell = \ell_b$ and $\boldsymbol{n}_{\min} = \min_{a_k \in B_{b,\ell}} \boldsymbol{n}_{b,\ell}(a_k)$
3: Receive the click feedback $\mathcal{C}_t = (\mathcal{C}_1^t, ..., \mathcal{C}_L^t)$
4: **for** $k = \mathbf{I}_b(1) : \mathbf{I}_b(2)$ **do**
5: $\quad$ **if** $\boldsymbol{n}_{b,\ell}(\mathbf{a}_k^t) = \boldsymbol{n}_{\min}$ **then**
6: $\quad\quad$ Set $\mathcal{C}_{b,\ell}(\mathbf{a}_k^t) = \mathcal{C}_{b,\ell}(\mathbf{a}_k^t) + \sum_{s=1}^L \mathcal{C}_s^t \mathbb{1}\{a_s = \mathbf{a}_k^t\}$ and $\boldsymbol{n}_{b,\ell}(\mathbf{a}_k^t) = \boldsymbol{n}_{b,\ell}(\mathbf{a}_k^t) + 1$

---

---

**Algorithm 7** UpdateBatch

---

1: **Input:** batch index $b$, time $t$
2: Set $\ell = \ell_b$
3: **if** $\min_{a_k \in B_{b,\ell}} \boldsymbol{n}_{b,\ell}(a_k) = \boldsymbol{n}_\ell$
4: $\quad$ **for all** $a_k \in B_{b,\ell}$ **do**
5: $\quad\quad$ Compute $U_{b,\ell}(a_k)$ and $L_{b,\ell}(a_k)$
6: $\quad$ Let $a_1, ..., a_{|B_{b,\ell}|}$ be any permutation of $B_{b,\ell}$ such that $L_{b,\ell}(a_1) \geq ... \geq L_{b,\ell}(a_{|B_{b,\ell}|})$
7: $\quad$ **for** $k = 1 : len(b)$ **do**
8: $\quad\quad$ Set $B_k^+ = \{a_1, ..., a_k\}$ and $B_k^- = B_{b,\ell} \backslash B_k^+$
9: $\quad$ **for** $k = 1 : len(b) - 1$ **do**
10: $\quad\quad$ **if** $L_{b,\ell}(a_k) > \max_{a_k \in B_k^-} U_{b,\ell}(a_k)$ **then**
11: $\quad\quad\quad$ Set $s = k$
12: $\quad$ **if** $s = 0$ and $|B_{b,\ell}| > len(b)$ **then**
13: $\quad\quad$ Set $B_{b,\ell+1} = \{a_k \in B_{b,\ell} : U_{b,\ell}(a_k) \geq L_{b,\ell}(a_{len(b)})\}$ and $\ell = \ell + 1$
14: $\quad$ **else if** $s > 0$ **then**
15: $\quad\quad$ Set $\mathbb{B} = \mathbb{B} \bigcup \{b_{\max} + 1, b_{\max} + 2\} \backslash \{b\}$, $B_{b_{\max}+1,0} = B_s^+$, $B_{b_{\max}+2,0} = B_s^-$, $\ell_{b_{\max}+1} = 0$
16: $\quad\quad$ $\ell_{b_{\max}+2} = 0$, $\mathbf{I}_{b_{\max}+1} = (\mathbf{I}_b(1), \mathbf{I}_b(1) + s - 1)$, $\mathbf{I}_{b_{\max}+2} = (\mathbf{I}_b(1) + s, \mathbf{I}_b(2))$, $b_{\max} = b_{\max} + 2$

---

After the CollectClicks section, the ranker would compute the KL-upper confidence bound and lower confidence bound (Garivier and Cappé, 2011; Zoghi et al., 2017) for every item in the batch, denote as $U_{b,\ell}(a_k)$ and $L_{b,\ell}(a_k)$

$$
\begin{aligned}
U_{b,\ell}(a_k) &= \arg\max_{q\in[\hat{\mathcal{C}}_{b,\ell}(a_k),1]} \{\boldsymbol{n}_\ell D_{KL}(\hat{\mathcal{C}}_{b,\ell}(a_k)\|q) \le \log(T) + 3\log\log(T)\} \\
L_{b,\ell}(a_k) &= \arg\min_{q\in[0,\hat{\mathcal{C}}_{b,\ell}(a_k)]} \{\boldsymbol{n}_\ell D_{KL}(\hat{\mathcal{C}}_{b,\ell}(a_k)\|q) \le \log(T) + 3\log\log(T)\}
\end{aligned}
\tag{26}
$$

where $D_{KL}$ represents the *Kullback-Leibler divergence* between Bernoulli random variables with means $p$ and $q$. In the UpdateBatch section, all the items in batch $B_{b,\ell_1}$ would be placed by order $a_1, ..., a_{|B_{b,\ell_1}|}$, where $L_{b,\ell_1}(a_1) \ge, ..., \ge L_{b,\ell_1}(a_{|B_{b,\ell_1}|})$. The BatchRank would compare the first $len(b) - 1$ item's lower confidence bound to the maximal upper confidence bound in $B_k^-$. If $L_{b,\ell_1}(a_k) > \max_{a_k\in B_k^-} U_{b,\ell_1}(a_k)$, the BatchRank would set $s = k$. Ones $s > 0$, the batch would spilt from position $s$ and the ranker derives sub-batches $B_{b+1,\ell_2}$ and $B_{b+2,\ell_3}$. Sub-batch $B_{b+1,\ell_2}$ contains $s$ items and the first $s$ positions in $\mathcal{R}_t$ and sub-batch $B_{b+2,\ell_3}$ contains $L - s$ items and positions from $s$ to $K$. The BatchRank would restart with stages $\ell_2 = 0$ and $\ell_3 = 0$ and sub-batches $B_{b+1,\ell_2}$ and $B_{b+2,\ell_3}$. The batches would recursively run and split until round $T$.

### E.2   Proof of Theorem 3

The proof of Theorem 3 relies on the following Lemma 3.

**Lemma 3.** *The attacker utilizes Algorithm 2 to manipulate the returned click feedback of the BatchRank. After $16L\log(T)$ rounds attack and the BatchRank begins its first split. The upper confidence bound and lower confidence bound of every non-target item satisfies $L_{b,\ell_1}(a_k) = 0$ and $U_{b,\ell_1}(a_k) = 1 - (T\log(T)^3)^{-1/\boldsymbol{n}_{\ell_1}}$. The lower confidence bound and the upper confidence bound of the target item are $L_{b,\ell_1}(\tilde{a}) = 1$ and $U_{b,\ell_1}(\tilde{a}) = 1$.*

*Proof of Lemma 3.* According to the introduction of BatchRank, the estimated click probability of an arbitrary item is written as (25) and $\mathcal{C}_{b,\ell_1}(a_k)$ is at most $16\log(T)$ in the first stage ($\ell_1 = 0$ and $\tilde{\Delta}_{\ell_1}^{-2} = 2^{2\ell_1} = 1$). Recall our attack Algorithm 2 returns $\tilde{\mathcal{C}}_k^t = 0$ when $a_k \ne \tilde{a}$ and $a_k \in \mathcal{R}_t$. Thus, the total collected click number of the non-target item is $\mathcal{C}_{b,\ell_1}(a_k) = 0$, and the estimated click probability is $\hat{\mathcal{C}}_{b,\ell_1}(a_k) = 0$.

We first introduce the definition of the KL-divergence

$$
D_{KL}(p\|q) = p\log(\frac{p}{q}) + (1-p)\log(\frac{1-p}{1-q}).
\tag{27}
$$

By convenience, we define $0\log(0) = 0\log(0/0) = 0$ and $x\log(x/0) = +\infty$ for $x > 0$ (Garivier and Cappé, 2011). With this knowledge, we can derive the upper confidence bound of the non-target item in stage $\ell_1$

$$
\begin{aligned}
U_{b,\ell_1}(a_k) &= \arg\max_{q\in[\hat{\mathcal{C}}_{b,\ell_1}(a_k),1]} \{\boldsymbol{n}_{\ell_1} D_{KL}(\hat{\mathcal{C}}_{b,\ell_1}(a_k)\|q) \le \log(T) + 3\log\log(T)\} \\
&= \arg\max_{q\in[0,1]} \{\boldsymbol{n}_{\ell_1}(0\log\frac{0}{q} + 1\log\frac{1}{1-q}) \le \log(T) + 3\log\log(T)\} \\
&= \arg\max_{q\in[0,1]} \{\boldsymbol{n}_{\ell_1}\log\frac{1}{1-q} \le \log(T) + 3\log\log(T)\}.
\end{aligned}
\tag{28}
$$

Apparently, when $q = 1$, $\log(T) + 3\log\log(T) \le \boldsymbol{n}_{\ell_1}\log(1/1-q) = +\infty$, hence $q$ should smaller than 1. When $\boldsymbol{n}_{\ell_1}\log(1/(1-q)) = \log(T) + 3\log\log(T)$, we have

$$
\begin{aligned}
\boldsymbol{n}_{\ell_1}\log(\frac{1}{1-q}) &= \log(T\log(T)^3) \\
\log(\frac{1}{1-q}) &= \log\left((T\log(T)^3)^{1/\boldsymbol{n}_{\ell_1}}\right) \\
\frac{1}{1-q} &= (T\log(T)^3)^{1/\boldsymbol{n}_{\ell_1}} \\
q &= 1 - (T\log(T)^3)^{-1/\boldsymbol{n}_{\ell_1}}.
\end{aligned}
\tag{29}
$$

Due to $T \log(T)^3 > 1$ and $\boldsymbol{n}_{\ell_1} > 0$, we can derive $0 < (T \log(T)^3)^{-1/\boldsymbol{n}_{\ell_1}} < 1$ and $0 < U_{b,\ell_1}(a_k) < 1$.

The lower confidence bound of the non-target item has

$$L_{b,\ell_1}(a_k) = \underset{q \in [0,0]}{\arg\min}\{\boldsymbol{n}_{\ell_1} D_{KL}(\hat{\mathcal{C}}_{b,\ell_1}(a_k) \| q) \leq \log(T) + 3\log\log(T)\} = 0. \tag{30}$$

Remember the attacker returns $\tilde{\mathcal{C}}_k^t = 1$ if $a_k = \tilde{a}$ and $a_k \in \mathcal{R}_t$. Thus, the total collected click number of target item is $\mathcal{C}_{b,\ell_1}(\tilde{a}) = 16\log(T)$ and $\hat{\mathcal{C}}_{b,\ell_1}(\tilde{a}) = 1$. We can further deduce the upper confidence bound of the target item as

$$U_{b,\ell_1}(\tilde{a}) = \underset{q \in [1,1]}{\arg\max}\{\boldsymbol{n}_{\ell_1} D_{KL}(\hat{\mathcal{C}}_{b,\ell_1}(\tilde{a}) \| q) \leq \log(T) + 3\log\log(T)\} = 1. \tag{31}$$

The lower confidence bound of the target item has

$$\begin{aligned} L_{b,\ell_1}(\tilde{a}) &= \underset{q \in [0,1]}{\arg\min}\{\boldsymbol{n}_{\ell_1} D_{KL}(\hat{\mathcal{C}}_{b,\ell_1}(\tilde{a}) \| q) \leq \log(T) + 3\log\log(T)\} = 1 \\ &= \underset{q \in [0,1]}{\arg\min}\{\boldsymbol{n}_{\ell_1}(1\log\frac{1}{q} + (1-1)\log\frac{1-1}{1-q}) \leq \log(T) + 3\log\log(T)\} \\ &= \underset{q \in [0,1]}{\arg\min}\{\boldsymbol{n}_{\ell_1}\log\frac{1}{q} \leq \log(T) + 3\log\log(T)\} \\ &= 1. \end{aligned} \tag{32}$$

Here finish the proof of Lemma 3. $\qquad \square$

*Proof of Theorem 3.* Consider the attacker implements attack Algorithm 2 with $T_1 = 16L\log(T)$. With the knowledge of Lemma 3, we can obtain when the BatchRank begins to split the first batch $B_{1,\ell_1} = \mathcal{D}$, the lower confidence bound of every non-target item satisfies $L_{b,\ell_1}(a_k) = 0$, and the lower confidence bound of the target item satisfies $L_{b,\ell_1}(\tilde{a}) = 1$. Therefore $\tilde{a}$ would be ranked at the first position because it has the highest lower confidence bound (line 6 in UpdateBatch). The BatchRank starts comparing $L_{b,\ell_1}(a_k)$ and $\max_{a_k \in B_k^-} U_{b,\ell_1}(a_k)$ for $k = 1$ to $K - 1$ (line 10 in UpdateBatch). Owing to $L_{b,\ell_1}(\tilde{a}) = 1 > U_{b,\ell_1}(a_k)$ and $L_{b,\ell_1}(a_k) < U_{b,\ell_1}(a_k)$, the split point is $s = 1$ (line 11 in Updatebatch). After the split action, the BatchRank would derive two sub-batches $B_{2,\ell_2} = \{\tilde{a}\}$ and $B_{3,\ell_3} = \mathcal{D} \backslash \tilde{a}$. Sub-batch $B_{2,\ell_2}$ contains the first position of $\mathcal{R}_t$ (i.e., $\mathbf{a}_1^t$) and $B_{3,\ell_3}$ contains the rest of the positions of $\mathcal{R}_t$ (i.e., $\mathbf{a}_2^t, ..., \mathbf{a}_K^t$). Sub-batch $B_{2,\ell_2}$ would not split until round $T$ because it only contains a position and an item. This implies after round $16L\log(T)$, the target item would always be placed at the first position of $\mathcal{R}_t$ until round $T$ is over, i.e., $N_T(\tilde{a}) \geq T - 16L\log(T)$. Due to the click number in each round being at most $K$, the cost in one round is at most $K$. Hence, the cost of Algorithm 2 can be bounded by $\mathcal{C}(T) \leq KT_1$.

Based on the above results, we conclude that Algorithm 2 can efficiently attack BatchRank when $T_1 = 16L\log(T)$. Here finish the proof of Theorem 3. $\qquad \square$

# F Proof of Theorem 4

## F.1 Introduction of TopRank

We here specifically illustrate details of the TopRank. The pseudo-code of the TopRank is provided.

The TopRank would begin with a blank graph $G_1 \subseteq [L]^2$. A directional edge $(a_j, a_i) \in G_t$ denotes the TopRank believes item $a_i$'s attractiveness is larger than item $a_j$. Let $\min_{G_t}(\mathcal{D} \backslash \bigcup_{c=1}^{d-1} \mathcal{P}_{tc}) = \{a_i \in \mathcal{D} \backslash \bigcup_{c=1}^{d-1} \mathcal{P}_{tc} : (a_i, a_j) \notin G_t \text{ for all } a_j \in \mathcal{D} \backslash \bigcup_{c=1}^{d-1} \mathcal{P}_{tc}\}$. The algorithm would begin from round 1 to round $T$. In each round, the TopRank would establish blocks $\mathcal{P}_{t1}, ..., \mathcal{P}_{td}$ via the graph $G_t$. Items in block $\mathcal{P}_{t1}$ would be placed randomly at the first $|\mathcal{P}_{t1}|$ positions in $\mathcal{R}_t$, and items in $\mathcal{P}_{t2}$ would be placed randomly at the next $|\mathcal{P}_{t2}|$ positions, and so on. In each round, after deriving click feedback $\mathcal{C}_t$, the TopRank would compute $U_{tij} = \mathcal{C}_i^t - \mathcal{C}_j^t$ if item $a_i$ and item $a_j$ are in the same block, otherwise, $U_{tij} = 0$. Afterward, the TopRank would

---

**Algorithm 8** The TopRank (Lattimore et al., 2018)

---

1: **Input:** Graph $G_1 = \emptyset$, round number $T$, $c = \frac{4\sqrt{2/\pi}}{\text{erf}(\sqrt{2})} \approx 3.43$

2: **for** $t = 1 : T$ **do**

3:      Set $d = 0$

4:      **while** $\mathcal{D} \backslash \bigcup_{c=1}^{d} \mathcal{P}_{tc} \neq \emptyset$ **do**

5:          Set $d = d + 1$

6:          Set $\mathcal{P}_{td} = \min_{G_t} \left( \mathcal{D} \backslash \bigcup_{c=1}^{d-1} \mathcal{P}_{tc} \right)$

7:      Choose $\mathcal{R}_t$ uniformly at random from $\mathcal{P}_{t1}, ..., \mathcal{P}_{td}$

8:      Observe click feedback $\mathcal{C}_t = (\mathcal{C}_1^t, ..., \mathcal{C}_L^t)$

9:      **for** $(i, j) \in [L]^2$ **do**

10:        **if** $a_i, a_j \in \mathcal{P}_{td}$ for some $d$ **then**

11:            Set $U_{tij} = \mathcal{C}_i^t - \mathcal{C}_j^t$

12:        **else**

13:            Set $U_{tij} = 0$

14:      Set $S_{tij} = \sum_{s=1}^{t} U_{tij}$ and $N_{tij} = \sum_{s=1}^{t} |U_{tij}|$

15:      Set $G_{t+1} = G_t \bigcup \left\{ (a_j, a_i) : S_{tij} \geq \sqrt{2N_{tij} \log(\frac{c}{\delta} \sqrt{N_{tij}})} \text{ and } N_{tij} > 0 \right\}$

---

compute $S_{tij} = \sum_{s=1}^{t} U_{sij}$ and $N_{tij} = \sum_{s=1}^{t} |U_{sij}|$ and establish edge $(a_j, a_i)$ if $S_{tij} \geq \sqrt{2N_{tij} \log(\frac{c}{\delta} \sqrt{N_{tij}})}$ and $N_{tij} > 0$. Without the attacker interference, the graph would not contain any cycle with probability at least $1 - \delta L^2$, if the graph contains at least one cycle the TopRank would behave randomly (Lattimore et al., 2018). Parameter $\delta$ would be set as $\delta = 1/T$.

### F.2 Proof of Theorem 4

The proof of Theorem 4 relies on the following lemmas.

**Lemma 4.** *Consider the TopRank is under the attack of Algorithm 2. Denotes $a_i = \tilde{a}$ as the target item and $a_j \neq \tilde{a}$ as non-target items. When $\sum_{t=1}^{T_1} \mathbb{1}\{\tilde{a} \in \mathcal{R}_t\} = 4\log(c/\delta)$, then $S_{T_1 ij} \geq \sqrt{2N_{T_1 ij} \log(\frac{c}{\delta} \sqrt{N_{T_1 ij}})}$ and $N_{T_1 ij} > 0$ are satisfied and edges from non-target items to target item (i.e., $(a_j, a_i)$, $a_j \neq a_i$) are established simultaneously.*

*Proof of Lemma 4.* Note that the TopRank sets $U_{tij} = \mathcal{C}_i^t - \mathcal{C}_j^t$ if $a_i, a_j \in \mathcal{P}_{td}$ for some $d$, otherwise, $U_{tij} = 0$. According to attack Algorithm 2, the TopRank would receive $\mathcal{C}_i^t = 1$ ($\mathcal{C}_i^t$ is generates by the target item) if $\tilde{a} \in \mathcal{R}_t$ and $\mathcal{C}_j^t = 0$ ($\mathcal{C}_j^t$ is generated by non-target items) when $t \leq T_1$. Based on this, we can derive

$$U_{tij} = \mathcal{C}_i^t - \mathcal{C}_j^t = 1, \ t \leq T_1, \ \tilde{a} \in \mathcal{R}_t, \ a_i, a_j \in \mathcal{P}_{td}. \tag{33}$$

Thus, when $\mathcal{P}_{t1} = \{\mathcal{D}\}$, we have

$$S_{tij} = \sum_{s=1}^{t} U_{tij} = N_{tij} = \sum_{s=1}^{t} |U_{tij}| = \sum_{s=1}^{t} \mathbb{1}\{\tilde{a} \in \mathcal{R}_t\}, \quad t \leq T_1. \tag{34}$$

In the light of (34) and line 15 of TopRank, if $\sum_{s=1}^{t} \mathbb{1}\{\tilde{a} \in \mathcal{R}_t\} \geq \sqrt{2 \sum_{s=1}^{t} \mathbb{1}\{\tilde{a} \in \mathcal{R}_t\} \log(\frac{c}{\delta} \sqrt{\sum_{s=1}^{t} \mathbb{1}\{\tilde{a} \in \mathcal{R}_t\}})}$, edges $(a_j, a_i)$ would establish. Utilizing the knowledge of the

elementary algebra, we have

$$\sum_{s=1}^{t} \mathbb{1}\{\tilde{a} \in \mathcal{R}_t\} \geq \sqrt{2 \sum_{s=1}^{t} \mathbb{1}\{\tilde{a} \in \mathcal{R}_t\} \log\left(\frac{c}{\delta} \sqrt{\sum_{s=1}^{t} \mathbb{1}\{\tilde{a} \in \mathcal{R}_t\}}\right)}$$

$$\left(\sum_{s=1}^{t} \mathbb{1}\{\tilde{a} \in \mathcal{R}_t\}\right)^2 \geq 2 \sum_{s=1}^{t} \mathbb{1}\{\tilde{a} \in \mathcal{R}_t\}\left(\log(\frac{c}{\delta}) + \log\left(\sqrt{\sum_{s=1}^{t} \mathbb{1}\{\tilde{a} \in \mathcal{R}_t\}}\right)\right)$$

$$\frac{1}{2} \sum_{s=1}^{t} \mathbb{1}\{\tilde{a} \in \mathcal{R}_t\} - \log\left(\sqrt{\sum_{s=1}^{t} \mathbb{1}\{\tilde{a} \in \mathcal{R}_t\}}\right) \geq \log(\frac{c}{\delta})$$

$$\sum_{s=1}^{t} \mathbb{1}\{\tilde{a} \in \mathcal{R}_t\} \geq 4 \log(\frac{c}{\delta}). \tag{35}$$

The second inequality holds because of $\sum_{s=1}^{t} \mathbb{1}\{\tilde{a} \in \mathcal{R}_t\} > 0$. The fourth inequality holds because of $(1/4)x > log(\sqrt{x})$ when $x > 0$. Thus, when $\sum_{s=1}^{t} \mathbb{1}\{\tilde{a} \in \mathcal{R}_t\} \geq 4\log(c/\delta)$ and $t \leq T_1$, edges $(a_j, a_i)$ would establish simultaneously. We here finish the proof of Lemma 4. $\qquad\square$

**Lemma 5.** *Suppose input $T_1 = \frac{4\log(c/\delta)}{\frac{K}{L} + (1-\sqrt{1+8K/L})/4}$, then with probability at least $1 - \delta/c$, the TopRank would achieve $\sum_{t=1}^{T_1} \mathbb{1}\{\tilde{a} \in \mathcal{R}_t\} > 4\log(c/\delta)$.*

*Proof of Lemma 5.* According to the previous discussion, we can separate $T_1$ into two periods $P_1$ and $P_2$ (i.e., $T_1 = P_1 + P_2$). In period one $G_t = \emptyset$ and in period two $G_t$ only contains edges from non-target items to the target item. Based on the TopRank property, in period one $P(\tilde{a} \in \mathcal{R}_t | t \leq P_1) = K/L$ and in period two $P(\tilde{a} \in \mathcal{R}_t | P_1 + 1 \leq t \leq T_1) = 1$. Define a Bernoulli distribution $X$ that satisfies $X = 1$ with probability $K/L$. With the help of the Hoeffding inequality, we can derive

$$P\left(\sum_{t=1}^{T_1} X_t - \frac{K}{L}T_1 \leq -aT_1\right) \leq e^{-T_1 a^2/2}. \tag{36}$$

Set $T_1 = \frac{4\log(c/\delta)}{\frac{K}{L} + (1-\sqrt{1+8K/L})/4}$ and $a = -(1-\sqrt{1+8K/L})/4$. We can derive

$$P\left(\sum_{t=1}^{T_1} X_t \leq 4\log(c/\delta)\right) \leq \frac{\delta}{c}. \tag{37}$$

Further derivation shows that

$$P\left(\sum_{t=1}^{T_1} X_t > 4\log(c/\delta)\right) > 1 - \frac{\delta}{c}. \tag{38}$$

Follows the definition of the TopRank, one has

$$P\left(\sum_{t=1}^{P_1} \mathbb{1}\{\tilde{a} \in \mathcal{R}_t\} + \sum_{t=P_1+1}^{T_1} \mathbb{1}\{\tilde{a} \in \mathcal{R}_t\} > 4\log(c/\delta)\right) = P\left(\sum_{t=1}^{P_1} \mathbb{1}\{\tilde{a} \in \mathcal{R}_t\} + P_2 > 4\log(c/\delta)\right)$$

$$\geq P\left(\sum_{t=1}^{T_1} X_t > 4\log(c/\delta)\right) \tag{39}$$

where the first equation holds because $\sum_{t=P_1+1}^{T_1} \mathbb{1}\{\tilde{a} \in \mathcal{R}_t\} = P_2$. The last inequality holds because $P_2 \geq \sum_{t=P_1+1}^{T_1} X_t$. Combining (38) and (39), we can finally get

$$P\left(\sum_{t=1}^{T_1} \mathbb{1}\{\tilde{a} \in \mathcal{R}_t\} \geq 4\log(c/\delta)\right) > 1 - \frac{\delta}{c} \tag{40}$$

when $T_1 = \frac{4\log(c/\delta)}{\frac{K}{L}+(1-\sqrt{1+8K/L})/4}$. Here finish the proof of Lemma 5. $\qquad\square$

**Lemma 6.** *If the attacker implements attack Algorithm 2 and $T_1 = \frac{4\log(c/\delta)}{\frac{K}{L}+(1-\sqrt{1+8K/L})/4}$, the graph $G_t$ would not contain any cycle with probability at least $1-(L^2+1/c)\delta$.*

*Proof of Lemma 6.* We here analyze our attack Algorithm 2 would not case $G_t$ contains any cycle with high probability if the input $T_1 = \frac{4\log(c/\delta)}{\frac{K}{L}+(1-\sqrt{1+8K/L})/4}$. Consider the attacker implementing our attack strategy from round 1 to round $T_1$. Define $a_i = \tilde{a}$ and $a_j \neq \tilde{a}$. The attacker frauds the TopRank to believe the target item $\tilde{a}$ is clicked $\sum_{t=1}^{T_1} \mathbb{1}\{\tilde{a} \in \mathcal{R}_t\}$ times and non-target items are clicked 0 time in $T_1$. After $S_{tij} \geq \sqrt{2N_{tij}\log(\frac{c}{\delta}\sqrt{N_{tij}})}$ and $N_{tij} > 0$ are satisfied, the edges would be established at the same time and $\tilde{a}$ would belong to the first block (line 6 in the TopRank and Lemma 4 and 6). Note that during $T_1$, the attacker sets $\mathcal{C}_j^t = 0$. Thus

$$
\begin{aligned}
N_{T_1 ji} &= \sum_{t=1}^{T_1} |U_{tji}| = \sum_{t=1}^{T_1} |\mathcal{C}_j^t - \mathcal{C}_i^t| = \sum_{t=1}^{T_1} \mathbb{1}\{\tilde{a} \in \mathcal{R}_t\} \\
S_{T_1 ji} &= \sum_{t=1}^{T_1} U_{tji} = \sum_{t=1}^{T_1} (\mathcal{C}_j^t - \mathcal{C}_i^t) = -\sum_{t=1}^{T_1} \mathbb{1}\{\tilde{a} \in \mathcal{R}_t\}.
\end{aligned}
\tag{41}
$$

Since $U_{tij}$ and $U_{tji}$ would be 0 after $t > T_1$ (line 9-13 in TopRank), we can obtain $S_{T_1 ji} = -\sum_{t=1}^{T_1} \mathbb{1}\{\tilde{a} \in \mathcal{R}_t\}$ and $N_{T_1 ji} = \sum_{t=1}^{T_1} \mathbb{1}\{\tilde{a} \in \mathcal{R}_t\}$ hold when $t > T_1$. This implies the directional edges from the target item to non-target items would never establish, i.e., $-\sum_{t=1}^{T_1} \mathbb{1}\{\tilde{a} \in \mathcal{R}_t\} < \sqrt{2\sum_{t=1}^{T_1} \mathbb{1}\{\tilde{a} \in \mathcal{R}_t\}\log(\frac{c}{\delta}\sqrt{\sum_{t=1}^{T_1} \mathbb{1}\{\tilde{a} \in \mathcal{R}_t\}})}$. Besides, due to the received click number from non-target items being 0 in $T_1$, the $S_{T_1}$ and $N_{T_1}$ between non-target items would be 0. This implies the manipulation of the attacker would not influence the TopRank judgment of the attractiveness between non-target items. In other words, the TopRank under Algorithm 2 attack can be considered as the TopRank interacts with item set $\mathcal{D}\backslash\tilde{a}$ in $T - T_1$ rounds.

According to the above discussion and Lemma 5, if $T_1 = \frac{4\log(c/\delta)}{\frac{K}{L}+(1-\sqrt{1+8K/L})/4}$, then $S_{T_1 ij} \geq \sqrt{2N_{T_1 ij}\log(\frac{c}{\delta}\sqrt{N_{T_1 ij}})}$ would satisfy with probability at least $1 - \delta/c$. Besides, from round $T_1 + 1$ to $T$, cycles would occur with probability at most $\delta L^2$. Thus graph $G_t$ would not contain cycles with probability at least $1-(L^2+1/c)\delta$ until $T$. $\qquad\square$

*Proof of Theorem 4.* Suppose the attacker implements attack Algorithm 2 with input value $T_1 = \frac{4\log(c/\delta)}{\frac{K}{L}+(1-\sqrt{1+8K/L})/4}$. Then, the TopRank would establish edges from non-target items to $\tilde{a}$ with probability at least $1 - \delta/c$ (According to Lemma 4 and Lemma 5). Based on the analysis in Lemma 6, the cycle would appear with probability at most $(L^2+1/c)\delta$ and the first block would only contain $\tilde{a}$ till $T$. That is to say, the target item in block $\mathcal{P}_{t1}$ would always be placed at the first positions after $T_1$ with probability at least $1-(L^2+1/c)\delta$. Following Algorithm 2, the attacker would only manipulate the returned click feedback for $T_1$ times. Thus the attack cost can be bounded by $\mathcal{C}(T) \leq KT_1$.

According to the above observation, we summarize that Algorithm 2 can efficiently attack TopRank when $T_1 = \frac{4\log(c/\delta)}{\frac{K}{L}+(1-\sqrt{1+8K/L})/4}$. Here finish the proof of Theorem 4. $\qquad\square$

