# OpenReview forum: "Adversarial Attacks on Online Learning to Rank with Stochastic Click Models"
_TMLR — Accepted by TMLR_

### Review · Reviewer_AunX · 2024-04-20

**Summary Of Contributions:**

This paper studies poisoning attacks against online learning to rank (OLTR) . The paper proposes GA and ATQ algorithms for list poisoning attack and click poisoning attack, respectively. The authors verify the effectiveness both theoretically and empirically.

**Audience:**

Yes

**Claims And Evidence:**

Yes

**Requested Changes:**

1. The authors should discuss the relationship between this paper and the concurrent work (Zuo et al. 2023) in the introduction.
2. The paper may provide more discussions on the points mentioned in the "Weakness".

**Strengths And Weaknesses:**

Strengths:

The paper is well-written. The stuided problem is very interesting. Though I didn't check all proofs, the theoretical results seems sound and reasonable.

Weakness:

1. My main concern is that the paper does not provide detailed comparison between the proposed methods and the concurrent work (Zuo et al. 2023). Since (Zuo et al. 2023) has been published in NeurIPS 2023 (whose audience highly overlaps with TMLR's audience), the authors need to show that some individuals will be still interested in the findings of this paper even if they have read (Zuo et al. 2023). Specifically, I think the paper needs to answer following questions:

- What's the main technical challenge of fooling the ranker to place the target item on top of the list instead of just in the list?
- Notice that (Zuo et al. 2023) studied click poisoning attacks against most popular OLTR methods. However, ATQ seems not suitable for attacking PBM-UCB and CascadeUCB. What is the main challenge of extending ATQ to attack PBM-UCB and CascadeUCB?
- Notice that BatchRank (Zoghi et al., 2017) and Toprank (Lattimore et al., 2018) only require $\chi(1)\ge\dots\ge\chi(K)$, while this paper assumes $\chi(1)>\dots>\chi(K)$. What is the limiting factor of studying the case where two positions have the same examination probability? Actually, methods in (Zuo et al. 2023) can handle this case.

2. ATQ requires to know the time horizon T, which is hard to obtain in real-world applications.
3. I'm surpurising that the theoretical results of ATQ are gap-independent. Could authors give some explanation for such results?
4. Why ATQ requires $O(KL\log(T))$ cost to attack against BatchRank, but only requires $O(L\log(T))$ cost to attack against TopRank?

---

> ### Author Response · Authors · 2024-05-13
> **Rebuttal to Reviewer AunX: Part 1**
>
> We thank the reviewer for the positive comments on our paper's significance and clarity, and the constructive questions to help further clarify our idea.
>
> **Q1:** Detailed comparison between the proposed methods and the concurrent work (Zuo et al. 2023).
>
> **A1:** Thank you for the question. In their work, [Zuo et al, 2023] introduced two click poisoning attack algorithm specifically design for CascadeUCB and PBMUCB. These algorithms rely on the knowledge of the target rankers and the underling click model. They also introduced an general attack algorithm based on click poisoning, which operates under the assumption that the attacker possesses knowledge regarding the feasible feedback space of the victim's click model (refer to the last paragraph of page 8 of [Zuo et al, 2023] for details). The authors acknowledged that "without this knowledge, ensuring valid post-attack feedback becomes impossible". The aforementioned discussion highlights the ineffectiveness of their attack algorithms (two single-target attack algorithms and a general attack algorithm) when the underling click model's information is absent. Additionally, we employ Example 2 to illustrate that click poisoning attacks are hardly adapted to different click models. To overcome the strong assumption that attacker knows the underling click model, we propose the list poisoning-based GA algorithm, which achieves our attack objective (place the target item on the top of the list for $T - o(T)$ times with $o(T)$ cost) without relying on any knowledge of the underlying click model.
>
> We have added this discussion into the related work section (see the last paragraph of the related work section).
>
> **Q2**: What's the main technical challenge of fooling the ranker to place the target item on top of the list instead of just in the list?
>
> **A2**: Thank you for the question. We here highlight the main challenges.
>
> 1. In the list poisoning attack section, the spirit of our GA is to ensure that the target item possesses the highest attractiveness. However, we discovered that in the cascade model, rankers can display any permutation of the optimal list and still attain a $0$ regret. This implies that cascade algorithms might not position the best item at the top of the list for $T-o(T)$ times. To show that our GA can successfully attack most of the existing CascadeUCB algorithms, we use novel analysis (shown in the proof of Theorem 2) to prove that CascadeUCB would place the most attractive item to the top of the list $T-o(T)$ times. This result helps us showing that our GA can fool the CascadeUCB to place the target item on the top of the list for $T-o(T)$ times with $o(T)$ cost.
>
> 2. In the list poisoning attack section, we introduce a new attack strategy named ATQ to boost the target item to the list's top. This strategy is built on the phase elimination property of general rankers like TopRank and BatchRank. We observe that these algorithms tend to "assume" that the most attractive item is the best during certain explorations, consistently placing it at the top. Consequently, our ATQ aims to fool TopRank and BatchRank that the target item possesses the highest attractiveness within a short period (i.e., $T_1$). We believe the insight of the ATQ is novel to the bandit community.
>
> We have added this discussion into the related work section.
>
> **Q3:** Notice that (Zuo et al. 2023) studied click poisoning attacks against most popular OLTR methods. However, ATQ seems not suitable for attacking PBM-UCB and CascadeUCB. What is the main challenge of extending ATQ to attack PBM-UCB and CascadeUCB?
>
> **A3:** Thanks for your question. The effect of ATQ relies on BatchRank and TopRank algorithms' phased elimination property. Specifically, by repeatedly attacking for a short period (i.e., $T_1$), the attacker can trick these algorithms into always believe the target item has the highest attractiveness. However, CascadeUCB and PBM-UCB belong to  UCB algorithms. When the attacker stops, the true click feedback of other items will be revealed to the algorithms over time, leading UCB algorithms to realize the targeted item isn't the item with the highest attractiveness.
>
> We have incorporated this discussion into Remark 8.

---

> ### Author Response · Authors · 2024-05-13
> **Rebuttal to Reviewer AunX: Part 2**
>
> **Q4:** Notice that BatchRank (Zoghi et al., 2017) and Toprank (Lattimore et al., 2018) only require $\chi(1)\ge\chi(2)\ge,...,\chi(K)$, while this paper assumes $\chi(1)>\chi(2)>,...,>\chi(K)$. What is the limiting factor of studying the case where two positions have the same examination probability? Actually, methods in (Zuo et al. 2023) can handle this case.
>
> **A4:** Thank you for your question. The reason [Zoghi et al, 2017; Lattimore et al, 2018] do not make this assumption is because they are primarily concerned with achieving sublinear regrets rather than the specific position of an item in list $\mathcal{R}$. Similarly, [Zuo et al, 2023] also disregard the position of the target item in $\mathcal{R}$, as their setting assumes the $\tilde{a} \in \mathcal{R}$ is an efficient attack. We make this assumption on the fact that if $\chi(1) = \chi(2)$, both $(a_1,a_2,a_3,...,a_K)$ and $(a_2,a_1,a_3,...,a_K)$ can achieve zero regret in the position-based model. This contradicts the assumption of our stochastic click model, which states that only $\mathcal{R}^* = (a_1,a_2,a_3,...,a_K)$ can achieve zero regret. More specifically, if $\chi(1) = \chi(2)$, then an algorithm can present $(a_2,a_1,a_3,...,a_K)$ for $T-o(T)$ to achieve a $o(T)$ regret, and GA can not achieve its attack goal due to $\tilde{a}$ has the highest attractiveness and it will be placed on the second position. Besides, it's important to note that we can alleviate the assumption to $\chi(1) > \chi(2) \geq \dots \geq \chi(K)$ and our GA can also achieve its attack goal in the position-based model. This is because we only focus on the position of the target item and disregard the positions of non-attractive items $\\{\eta_k\\}_{k=1}^{K-1}$.
>
> Furthermore, suppose $\chi(1)>\chi(2)$ is highly plausible because in real-world scenarios, it is impossible for $\chi(i) = \chi(j)$ to occur, as users inherently exhibit biases towards distinct positions.
>
> We have incorporated this discussion into the Remark 4.
>
> **Q5:** ATQ requires to know the time horizon $T$, which is hard to obtain in real-world applications.
>
> **A5:** Thank you for your comment. It's worth noting that both TopRank and BatchRank operate under the assumption that the time horizon $T$ is known, making the scenario of the attacker possessing this knowledge quite plausible. Exploring new ATQ, BatchRank and TopRank that do not rely on the knowledge of $T$ is an interesting topic for future research.
>
> **Q6:** I'm surpurising that the theoretical results of ATQ are gap-independent. Could authors give some explanation for such results?
>
> **A6:** Thank you for the question. The reason the cost upper bound of the ATQ isn't gap-dependent is due to 1) the approach taken during the attack, and 2) the structure of the BatchRank and TopRank. The TopRank and BatchRank algorithms are classified as phase elimination algorithms. In the proofs of Theorems $3$ and $4$, we demonstrate that we can achieve the attack objective by maximize the click feedback of the target item and minimize the click feedback of the non-target items during the first elimination phase (see ATQ pseudo code). Through analysis, we determine that the length of the first elimination phase for BatchRank and TopRank is respectively bounded by $O\big(L\log(T)\big)$ and $O\big(\frac{L}{K}\log(T)\big)$, which is unrelated to the attractiveness gap. Besides, due to TopRank and BatchRank will randomly explore the whole item set in the first phase, we suppose the attack cost per round is $K$ due to the attacker should modify at most $K$ click feedback in each round. Accordingly, the finial cost upper bound do not depends to the attractiveness gaps.
>
> We have added this discussion into Remark 6.
>
> **Q7:** Why ATQ requires $KL\log(T)$
>  cost to attack against BatchRank, but only requires $L\log(T)$ cost to attack against TopRank?
>
> **A7:** Thank you for the question, we believe this difference is owing to  TopRank can theoretically outperform BatchRank [Lattimore et. al., 2018]. Specifically, TopRank completes its first exploration phase (decision-making and ranking the target item on the top of the list) in $\frac{L}{K}\log(T)$ time, whereas BatchRank requires $L\log(T)$ for the same tasks. As per the ATQ strategy, this allows for a smaller $T_1$ to be utilized in attacking TopRank compared to BatchRank.
>
> We have added this discussion into Remark 6.
>
> Thank you again for these helpful questions, comments and suggestions.

---

> ### Author Response · Authors · 2024-05-13
> **Rebuttal to Reviewer AunX: Part 3**
>
> [1] Lattimore, T., Kveton, B., Li, S., & Szepesvari, C. (2018). TopRank: A practical algorithm for online stochastic ranking. Neural Information Processing Systems.
> [2] Zoghi, M., Tunys, T., Ghavamzadeh, M., Kveton, B., Szepesvari, C., & Wen, Z. (2017). Online Learning to Rank in Stochastic Click Models. International Conference on Machine Learning.
> [3] Zuo, J., Zhang, Z., Wang, Z., Li, S., Hajiesmaili, M.H., & Wierman, A. (2023). Adversarial Attacks on Online Learning to Rank with Click Feedback. ArXiv, abs/2305.17071.

---

> ### Author Response · Authors · 2024-08-05
> **Response to Reviewer Aunx**
>
> Thank you for the interesting question. We believe it is challenging for ATQ to employ the doubling trick to attack different ranking algorithms. For example, if the attack target is TopRank, the attack does not directly depend on $T$. As stated in Theorem 4, we need to set $T_1 = \frac{4\log(c/\delta)}{\frac{K}{L} + (1 - \sqrt{1+8K/L})/4}$, which depends on the parameters $\delta$ and $c$. In [Lattimore et al., 2018], they can set $\delta = 1/(T^a)$ ($a > 1$) and achieved a regret upper bound similar to when $\delta = 1/T$. Therefore, if the ATQ does not know how the TopRank selects its $\delta$ and $c$, it cannot successfully attack TopRank even if it knows the time horizon $T$.

---

### Review · Reviewer_UXbV · 2024-04-30

**Summary Of Contributions:**

The paper studies adversarial attacks in a setting of “online learning to rank”.

Base problem:

The goal of the ranker in this problem is to present an ordered tuple of some larger set of items to a user to maximize the probability that the user clicks. Their problem setting is fairly general: they only make the assumption that click probability at each round is independent of previous rounds. They consider an online setting where the ranker presents a new ranking at each round, gradually improving over time. To achieve the standard goal of sublinear regret (aka be a no-regret algorithm), in this particular setting, the algorithm must output the unique optimal ranking sufficiently often (i.e. T - o(T) steps). This property winds up being important for making the adversarial attack work.

Attack model:

The authors then consider attackers who want to manipulate such a no-regret algorithm.

They present two reasonable models of what an attacker can do — click poisoning attacks (motivated by the realistic problem of click fraud) and list poisoning attacks (motivated by an example where an entity could create a lot of fake items to manipulate outcomes). While stylized these seem relatively reasonable and convincing (the click model more so). Applying both of these attacks imposes a cost per round they have to be done.

The goal of the attacker is to mislead the no-regret algorithm into ranking some chosen item first, while making sure their own cost grows sublinearly.

Concrete attacks:

- Generalized attack under list poisoning model: the authors present an attack that can work for any no-regret ranker, by essentially deceiving it into thinking that a given list is the top, and so getting it to output that list many times.
- Attack-then-quit strategy: this is a strategy for the click poisoning attacker model, which can be shown to work on two popular ranking strategies (BatchRank and TopRank).

The authors run experiments showing these attacks work, both on synthetic data (similar setup to previous literature) and the popular MovieLens dataset.

**Audience:**

Yes

**Broader Impact Concerns:**

Although the topic of better adversarial attack strategies is one that might seem troubling from a broader impact perspective, this paper is mainly theoretically focused, so describing the algorithms and releasing code is likely not dangerous.

**Claims And Evidence:**

Yes

**Requested Changes:**

I have no requested changes -- the submission seems adequate as is.

**Strengths And Weaknesses:**

Strengths:

- A decently realistic model, though stylized.
- The algorithms are simple and it is obviously they could be easily implemented. It seems that costs are “realistic” (i.e. one doesn’t have to completely corrupt the entire signal for the ranker).
- Formal proofs of manipulability for popular LTR models.
- There is a clear connection to the types of online LTR models used in the literature. And one should definitely expect attackers to try to manipulate LTR models in the real world.

Weaknesses:
- As with many adversarial attack papers, it is sort of “obvious” that having strong power to steer rewards or tamper with actions could allow one to steer an algorithm’s behavior to bad places, if it is not designed to defend against this. That said, simplicity and lack of novelty don’t make these results bad or uninteresting — it is good that they have been worked out, and in particular that the costs faced by the attacker trying to run such an attack are clearly understood here.

---

> ### Author Response · Authors · 2024-05-13
> **Rebuttal to Reviewer UXbV**
>
> The authors express their heartfelt gratitude to the reviewer for the positive comment of the paper's novelty, significance and clarity.

---

### Review · Reviewer_yCyz · 2024-07-20

**Summary Of Contributions:**

This paper studies two possible attack scenarios in online learning to rank (OLTR): click poisoning attacks and list poisoning attacks. Their derivation demonstrates that the existing OLTR methods are very vulnerable to these attacks: the attacks can efficiently mislead the OLTR method to pick the target unwanted item as a top ranked item.

**Audience:**

Yes

**Claims And Evidence:**

Yes

**Requested Changes:**

Please address the issues mentioned in the weakness section.

**Strengths And Weaknesses:**

Strength:

The introduction of the attacks into the OLTR scenario is interesting.

Weakness:

The main issues of this paper are (1) the unclear description of the motivation, and (2) imprecise math notations and language.

For the motivation, while proposing the attacks into OLTR is interesting, besides deepen the understanding on the vulnerability and the sensitivity of OLTR methods, the authors need to provide more concrete description on why they consider these two specific attacks, and why it is essential to consider these attacks. For list poisoning attack, although Example 1 tries to provide a real scenario of the attack, it is not clear that how the attacker is able to insert the attack between the communication of the buyer and the website. The last sentence "When the e-commerce platform interacts with the user, the attacker can implement based on its uploaded items" needs more details. For click poisoning attack, the authors also need to suggest one practical scenario on how the attacker is able to conduct the attack.

For the writing, there are numerous imprecise math notations and grammar which impacts the quality of the paper. For math notations, while the list in the appendix is clear, the description in the main paper is not clear. For example, in equation (1), why we use $a_s$ rather than $a_k$ in the formula? Why $v(\mathcal{R},\textbb{a}_k^t,k)$ rather than $v(\mathcal{R},a_s,k)$? Also, on the line below equation (1), the $a_1,\ldots,a_L$ in $\mathcal{R}^*$ seems to represents a different thing compared to how they are introduced in the paragraph above equation (1).

In terms of the grammar issues, the first sentence of Assumption 1 has grammar (change "due to" to "since" or "because"). Remark 1 "correlation" should be "relationship" (correlation is a term with some specific math definition). Paragraph "List poisoning attack", the sentence "We suppose the attacker does not need to know the actual..." is incorrect in its logic: Please replace "does not need to know" with "does not know". The last sentence of Example 1, there should be a noun following the word "implement".

---

> ### Author Response · Authors · 2024-07-27
> **Rebuttal to Reviewer yCyz**
>
> **Q1:** The authors need to provide more concrete description on why they consider these two specific attacks, and why it is essential to consider these attacks. For list poisoning attack, although Example 1 tries to provide a real scenario of the attack, it is not clear that how the attacker is able to insert the attack between the communication of the buyer and the website. The last sentence "When the e-commerce platform interacts with the user, the attacker can implement based on its uploaded items" needs more details. For click poisoning attack, the authors also need to suggest one practical scenario on how the attacker is able to conduct the attack.
>
> **R1:** Thank you for your valuable suggestion.
>
> A list poisoning attack can be executed through malware installed as a browser extension, which manipulates the ranking list on the web page locally. This type of attack—where the manipulation targets the actions rather than the feedback—has also been studied in bandit and reinforcement learning scenarios by [Liu et al., 2020; 2021; 2022].
>
> Click poisoning attacks can also be executed through malware installed as a browser extension, which manipulates the click feedback signal locally before uploading it to the server (e.g., the OLTR algorithm). Such attacks were previously studied by [Golrezaei et al., 2020].
>
> We have improved the motivation section of our paper based on the above discussion.
>
> **Q2:** For the writing, there are numerous imprecise math notations and grammar which impacts the quality of the paper. For math notations, while the list in the appendix is clear, the description in the main paper is not clear. For example, in equation (1), 1) why we use $a_s$ rather than $a_k$ in the formula? 2) Why $v(\mathcal{R},\alpha^t_s,k)$ rather than $v(\mathcal{R},a_s,k)$? 3) Also, on the line below equation (1), the $a_1,...,a_L$ in $\mathcal{R}^*$ seems to represents a different thing compared to how they are introduced in the paragraph above equation (1).
>
> **R2:** Thank you for your comments and questions. We apologize for any confusion caused by the unclear definition in our paper.
>
> Eq (1) is writtern as $P(\mathcal{C}^t_s\ \vert\ \mathcal{R}_t = \mathcal{R}, \alpha_k^t = a_s) = v(\mathcal{R},\alpha_k^t,k)$.
>
> 1) In Eq (1), $k$ represents the $k$-th position of the list $\mathcal{R}_t = \mathcal{R}$. The notation $a_s = \alpha_k^t$ denotes that the $s$-th most attractive item is placed in the $k$-th position of the list $\mathcal{R}_t = \mathcal{R}$. Note that we use $a_s$ to denote an arbitrary item in the item set $\mathcal{D}$, and it does not necessarily need to be $a_k$ (i.e., the $k$-th most attractive item).
>
> 2) The notation $v(\mathcal{R}_t, \alpha_k^t, k)$ denotes the click probability of the item at the $k$-th position in round $t$, so we utilize $\alpha_k^t$ (denoted as the $k$-th item in $\mathcal{R}_t$) instead of $a_s$ to emphasize the position $k$.
>
> 3) The notation $\mathcal{R}^* = (a_1, \ldots, a_K)$ denotes the optimal item list containing the top $K$ most attractive items. In Section 2.1 (i.e., the paragraph preceding Eq (1)), we define $\alpha(a_1) > \alpha(a_2) > \cdots > \alpha(a_L)$ ($\alpha(a_k)$ represents the attractiveness of item $a_k$), which corresponds to the definition of $\mathcal{R}^*$.
>
> We have improved the clarity of our paper based on this discussion.
>
> **Q3:** In terms of the grammar issues, the first sentence of Assumption 1 has grammar (change "due to" to "since" or "because"). Remark 1 "correlation" should be "relationship" (correlation is a term with some specific math definition). Paragraph "List poisoning attack", the sentence "We suppose the attacker does not need to know the actual..." is incorrect in its logic: Please replace "does not need to know" with "does not know". The last sentence of Example 1, there should be a noun following the word "implement".
>
> **R3:** Thank you for your valuable suggestions. We have carefully revised the paper based on your suggestions.
>
> Once again, the authors sincerely thank the reviewer yCyz for these helpful comments and suggestions.
>
> **Reference:**
>
> [1] Liu G, Lai L. Action-manipulation attacks against stochastic bandits: Attacks and defense[J]. IEEE Transactions on Signal Processing, 2020, 68: 5152-5165.
> [2] Liu G, Lai L. Provably efficient black-box action poisoning attacks against reinforcement learning[J]. Advances in Neural Information Processing Systems, 2021, 34: 12400-12410.
> [3] Liu G, Lai L. Action Poisoning Attacks on Linear Contextual Bandits[J]. Transactions on Machine Learning Research, 2022.
> [4] Golrezaei N, Manshadi V, Schneider J, et al. Learning product rankings robust to fake users[C]//Proceedings of the 22nd ACM Conference on Economics and Computation. 2021: 560-561.

---

### Decision · Action_Editor_nuSG · 2024-09-11

**Recommendation:** Accept as is

**Comment:**

The reviewers originally raised concerns about not enough coverage of the related work, about clarity and notation, but these were all well addressed in the rebuttal and the revision, and they all recommended acceptance.

**Audience:**

Yes.

**Claims And Evidence:**

Yes.